# Effective remediation programs for vulnerable students to overcome learning loss

**Madelon Jacobs**[1]*, **Carla Haelermans**[1,2,3], **Martijn Meeter**[4,5]

**1** Research Centre for Education and the Labour Market (ROA), School of Business and Economics, Maastricht University, the Netherlands, **2** Netherlands Initiative for Education Research (NRO), The Hague, the Netherlands, **3** National Education Lab AI (NOLAI), Radboud University, Nijmegen, the Netherlands, **4** Vrije Universiteit Amsterdam, Amsterdam, the Netherlands, **5** Research Institute LEARN!, Amsterdam, the Netherlands

* mce.jacobs@maastrichtuniversity.nl

## Abstract

In March 2020, the world was forced to cope with the COVID-19 pandemic, and as a result, schools in many countries closed to minimize the spread of the coronavirus. Students were required to shift to online learning from home. Not long after that, research showed that students in the first half-year of the COVID-19 pandemic experienced extensive learning losses compared to peers from previous cohorts. An important question is whether these losses can be remediated or lead to permanently lower attainment for a generation of students. Here, we study the effectiveness of a nationwide attempt at remediation in the Netherlands, where the government provided funds for schools to proactively minimize the detrimental effects of the COVID-19 pandemic. Unique standardized test score data of over 66,000 Dutch primary school students were combined with information about the various remediation programs employed uniquely in each school to investigate to what extent remediation occurred. Applying a Difference-in-Differences design, we find an overall statistically significant and meaningful increase in the achievement of vulnerable students after participation in remediation programs (+ 0.05 SD). Participating students had a higher increase in test scores compared to non-participants within the same school, and the inequality between the latter and former has slightly been reduced (~10%). Remediation programs offering individual support to students or in small groups, and with a sole emphasis on cognitive skill development, were most successful.

## Introduction

The start of the COVID-19 pandemic in March 2020 forced countries around the globe to take rigorous measures to decrease the spread of the coronavirus. One of these measures was closing schools, and, in some countries, education shifted to online distance learning, whereas in other countries, students received no education

**Data availability statement:** This section contains information on the data collection for the study "The Effects of Remediation Programs during the COVID-19 Pandemic on the Achievements of Vulnerable Students". The study is based on data from the Netherlands Cohort Study on Education (NCO in Dutch) and information about remediation programs sourced from schools, including participation lists and content details. Access to the NCO data is only possible via the Remote Access (RA) infrastructure of Statistics Netherlands. The data cannot be shared publicly because of privacy restrictions and cannot be exported from the secure virtual environment at Statistics Netherlands. However, all data are available at Statistics Netherlands, and researchers can request data access provided they work at an institution with which Statistics Netherlands has agreements. Requests for data access can be sent to info@nationaalco-hortonderzoek.nl. A detailed description of the data and the access procedure can be found in Haelermans, C., T. Huijgen, M. Jacobs, M. Levels, R. van der Velden, L. van Vugt and S. van Wetten (2020). Using Data to Advance Educational Research, Policy and Practice: Design, Content and Research Potential of the Netherlands Cohort Study on Education. European Sociological Review, 36(4), 643-662, https://doi.org/10.1093/esr/jcaa027. However, data on remediation programs are not openly requestable for researchers. Data obtained from funding recipients and participation lists obtained from schools are confidential and cannot be shared due to legal and ethical restrictions. Researchers seeking access for replication purposes may contact the authors and can create a project at Statistics Netherlands (provided they work at an institution with which Statistics Netherlands has agreements). The authors are allowed to provide the data on the remediation programs under strict conditions within other projects at Statistics Netherlands.

**Funding:** This research was supported by the Netherlands Initiative for Education Research (NRO), project number 40.5.20937.001. The funders had no role in study design, data collection and analyses, decision to publish, or preparation of the manuscript.

**Competing interests:** The authors have declared that no competing interests exist.

during the COVID-19 pandemic at all [1]. Consequently, it has created a severe crisis in education and learning [2], where students spent between one week to more than one school year at home [3]. Research showed significant learning losses after the school closures compared to students from previous cohorts in countries worldwide [4–11]. Vulnerable students faced disproportionate learning losses after the school closures, causing inequalities to rise between students [12,13]. The literature uses many terms to describe the detrimental effects of school closures on students' achievements, including learning delays, learning growth, learning loss, student achievements, and educational outcomes. This paper uses the term learning loss.

With the apparent learning losses and the affected students in countries world-wide, reopening schools and offering remediation was urgent. Half of all countries worldwide increased their education budget in 2020 and 2021 compared to pre-COVID-19 years [14]. Around 40% of all nations made various efforts to counteract the learning loss, from extending the academic year to prioritizing specific skills over other learning areas usually taught in school. Furthermore, many schools implemented programs specifically for the most vulnerable and disadvantaged students. These programs aimed to assist students who lagged behind in school, enabling them to catch up, counteract learning losses, and avoid grade retention.

In a quick attempt to counteract the learning losses, the Dutch Ministry of Education, Culture and Science provided funds to schools for remediation programs after the first half-year of the COVID-19 pandemic. In this paper, we focus on the introduction of these well-funded remediation programs in primary education in the Netherlands. Using the exceptional nationwide database of standardized test scores in the Netherlands to assess learning losses, these data can provide insight into the remediation efforts provided by these programs. We refer to these programs, set up to tackle the COVID-19 learning loss, as "remediation programs", as these are not clean individual interventions set up in experimental settings but are more fuzzy and often consist of many elements.

This research explores the characteristics of students who participated in remediation programs, identifying the criteria for their selection (research aim 1). Furthermore, it examines the relationship between participation in these programs and student achievements (research aim 2). Additionally, the study aims to explore the diversity in characteristics of remediation programs to identify the most effective characteristics in enhancing student achievements (research aim 3). The remainder of this paper shows that vulnerable students are rightfully selected for remediation programs and that, on average, these programs increase students' achievements.

## Relevant literature

After the first school closures due to the COVID-19 pandemic, many studies looked at students' learning losses in school, and several review studies are now available [4,8,11]. These review studies show that in almost all countries around the world, students learned less during COVID-19. For example, Betthäuser, Bach-Mortensen [11] show that learning losses occurred across 15 countries, based on 42 studies, with an average learning loss of 0.14 SD. Furthermore, they show that learning losses are

more prominent among young students and students from lower socio-economic backgrounds [8,15]. Besides the learning losses in mathematics and reading, students also struggled in other areas related to education, such as their non-cognitive skills or their well-being [16–18]. We refer to the cited review studies for more discussion of learning loss; below, we focus on studies that looked at potential solutions for remediating cognitive and/or non-cognitive skills.

With the knowledge of the learning losses occurring after the school closures, initiatives have been taken to make up for them. Even before the COVID-19 pandemic, the effectiveness of educational interventions was an important research topic. Numerous studies have examined effective interventions to improve skills such as reading, mathematics, and other cognitive and non-cognitive skills [15,19–35]. Many types of interventions have been developed to assist struggling students with additional support, time, and instruction. These type of interventions often have several important aspects in common. The first common aspect is that additional instruction is more tailored to the specific needs of struggling students. If classroom instructions are too challenging, students will not develop effectively. Hence, the additional instructional time should be better adjusted to the needs of the students. This aligns, for example, with Vygotsky's theory on the zone of proximal development [36], where students require a specific level of difficulty to perform optimally. The second common aspect is that additional instruction or support is often provided in smaller groups. This allows students better opportunities to receive (more) individual feedback and improves their engagement. This corresponds with Slavin's findings on small-group instruction [37]. Thirdly, additional instruction could also be defined as more guidance, where struggling students receive additional guidance compared to peers in the classroom. This is in harmony with Sweller's cognitive load theory [38], which states that students learn more effectively when their cognitive load is minimised through structured, step-by-step guidance. These three common aspects occur in many, if not most, of the interventions focused on remediation. Below, we will summarize the interventions that are most applicable to the case of the Netherlands.

One of the most effective intervention, is *one-to-one tutoring.* In this intervention, a student receives individual support outside the classroom. The student receives extensive individual support specifically targeting particular areas, such as mathematics or reading. The average effect size is estimated to be around 0.40 SD increase in attainment or roughly six months of additional progress [22]. On average, one-to-one tutoring seems more effective for reading (+0.49 SD) than mathematics (+0.14 SD) [22].

Besides one-to-one tutoring, *small-scale tutoring* is an often used intervention. This intervention may be preferred as one-to-one tutoring can be expensive. Small-scale tutoring teaches several students simultaneously, often between two and five students per group. Groups larger than six or seven students see a reduction in effectiveness [24]. The group setting allows students to learn from each other and the teacher. In this type of intervention, the average effect size is around 0.31 SD increase in test scores [24,27]. Similar to one-to-one tutoring, the intervention seems to be slightly more effective for reading (+0.31 SD) compared to mathematics (+0.23 SD) [24].

*Smaller classrooms* are also an option to provide more customized education. However, it is argued to be a costly intervention for relatively smaller effect sizes [25]. Average effect sizes are considered low, with an increase of around 0.14 SD, or around 2 months of additional progress. Smaller classrooms also seem more effective for reading (+0.14 SD) than mathematics (+0.08 SD).

Teaching students effective *learning strategies* could further improve educational performance for struggling students. Research indicates that metacognition improves performance, often defined as learning to learn [39]. Teaching students how to plan, monitor, and evaluate their own learning increases their ability to retain information and apply knowledge. These interventions equip students with tools to learn more effectively on their own. Research indicates that the average effect sizes range from 0.62 to 0.69 SD, corresponding to about 8 months of progress. Metacognitive strategies have been particularly effective for mathematics [40].

*Individualized instruction*, or differentiation, involves providing students personalized tasks and support, often while using educational technology. It is based on the idea that all students have unique needs, and thus, a customized approach, often regarding the pace of learning, will increase educational performances. Various strategies of

individualized instruction have been studied, often for mathematics, where students can work through individual sets of tasks, often independently. Average effect sizes range from approximately 0.19 to 0.26 SD, corresponding to about three months of progress [41].

*Extended school days*, where students stay at school for longer, increase students' educational achievements. If used effectively, this strategy provides more opportunities for direct instruction and practice after school [42]. Longer school days allow teachers more time to assess student progress and provide additional instruction. Without structured guidance, however, extending the school day could have no effect [43], where it may result in student fatigue and diminishing returns [42]. The average effect sizes of this intervention range from approximately 0.19 to 0.26 SD for both mathematics and reading in primary education [44].

*Summer schools* are another type of intervention that extends the official school year. They select a small group of students to receive additional teaching and instructions over the summer break [23,28,29,35]. The average effect size is around 0.23 SD, or 3 months of additional progress [23]. The effects appear larger for reading (0.23 SD) than for mathematic progress (0.14 SD).

Some types of interventions are targeted at all students within the classroom and are meant to increase the learning and support for all students, such as improving *instructional quality* in the regular classroom and the *professional development of teachers*. Successful programs of this type are characterized by their emphasis on student engagement and motivation strategies [34], which are expected to enhance student learning outcomes. These interventions are potential solutions for providing additional support to low-achieving students, without organizing additional programs for these students, with average effect sizes ranging around 0.33 SD for mathematics [33] or 0.56 SD for reading [27].

*Improving resources and study material* is another type of intervention that can be used for struggling students [30–32]. Offering additional resources to schools can include increasing the number of teachers [30], but often refers to implementing adaptive online programs for specific subjects [19]. Research by Ma, Adesope [32] shows adaptive online programs improve achievement in math and reading, with effect sizes ranging from 0.36 to 0.57 SD compared to traditional instruction. However, Steenbergen-Hu and Cooper [31] suggest these programs are more effective for students with higher prior knowledge, which may limit their effectiveness in the COVID-19 setting.

It is important to note that the success of the interventions discussed above is contingent upon the quality of their implementation and the expertise of the teacher/tutor(s) involved. Therefore, it is more accurate to consider these interventions as "promising" rather than guaranteed solutions. Their effectiveness may not be as pronounced as demonstrated in prior studies, and the potential for variation in outcomes should be taken into account. We expect to find lower effects compared to experimental interventions.

## Setting in the Netherlands

In the Netherlands, students experienced eight weeks of distance learning due to school closures related to COVID-19, with teaching functioning at a 30% efficiency compared to traditional in-school teaching in the spring of 2020 [10]. The first official school closure lasted for eight weeks, including two weeks of holidays, so students missed six weeks of school. After the schools reopened, students had four weeks of half-time learning at school and half-time distance learning at home, so they missed an additional two weeks of learning at school. In total, therefore, students missed eight weeks of school.

Previous research in the Netherlands showed significant learning losses and rising inequalities among students [5,7] after the school closure from March 16th, 2020, to May 10th, 2020 [45]. The results showed that students had, on average, a learning loss around 0.08 SD, while students from lower-educated parents faced 60% larger losses [5]. Other research in the Netherlands showed that after a full year of the COVID-19 pandemic, including two school closures, average learning losses were 0.17 SD for reading and 0.12 SD for mathematics [7]. Furthermore, based on parental education and parental income, students from disadvantaged backgrounds faced larger losses on top of already existing inequalities.

To quickly counteract the learning losses, the Dutch Ministry of Education, Culture and Science provided funds (around 3.4 billion euros, which is around 8% of the yearly education budget, or 0.3% of GDP) to schools for remediation programs after the first half-year of the COVID-19 pandemic. When schools fulfilled the specified criteria for the program, they were eligible to seek financial assistance from the Ministry. These conditions included a minimum of 25 hours of remediation per student, the involvement of the lowest-performing students, and the ability to request funds for up to 10% of the total student population in the school. Most schools received funds for 10% of their students. However, disadvantaged schools received funds for 20% of their students. This was determined based on the school-disadvantage score, which expresses expected educational disadvantages at the school level. Furthermore, the programs had to be carried out in the school year 2020/2021. Besides these criteria, schools were free to design the remediation program as they deemed necessary to counteract the learning losses among their students [19]. Schools could apply for funding through several calls, the first call was between the 2nd and 21st of June 2020, and the second was between the 18th of August and the 18th of September 2020.

The Netherlands is an ideal case to assess the impact of remediation programs on students' achievement in primary education. The data infrastructure allows looking at educational trajectories and achievements using longitudinal administrative data from the Netherlands Cohort Study on Education [46]. National standardized test scores in comprehensive reading and mathematics measure students' achievements twice a year. Moreover, detailed information regarding the content and participants of the remediation programs, as well as students' background information, is available. Combined with register data from Statistics Netherlands, unique, high-quality data on 66,439 students is available.

## Data and methods

### Data

This study utilizes administrative data from the Netherlands Cohort Study on Education [46], consisting of student and school characteristics. Furthermore, we have information on standardized test scores throughout primary education from grades one to six retrieved from student monitoring systems. Additionally, unique student identification numbers allow us to combine these test scores with information regarding data on remediation programs.

These data on remediation programs consist of participation lists of all students, as schools were asked to provide participation lists, allowing us to identify which students took part in the remediation programs. Additionally, data on the characteristics of the remediation programs is available, retrieved from the funding application questionnaire filled in by the schools. We received ethical approval for this study under VCWE#2019-147. See S1 Appendix for more information on the data collection regarding remediation programs.

We start with a sample of roughly 500,000 unique students with test score data in school years 2019/2020 and 2020/2021. After combining this with information on participating and non-participating students, the sample consists of 140,000 unique students. Students with missing background information (N = ~500), as well as students who did not take a test in both 2019/2020 (before the program) and 2020/2021 (after the program) were removed from the sample. The remaining sample consisted of 66,439 students across 456 schools in the Netherlands. See S2 Appendix for more information about the sample's representativeness.

### Background characteristics

Student demographics include sex, migration background, parental education attainment, parental income level, household type, and parental labor market position. Table 1 presents descriptive statistics of student demographics split between participating and non-participating students. Furthermore, school-level data such as the demographic composition of the student population and school resource information, such as the school's denomination and urbanization-level and the school's disadvantage score are available (see S3 Appendix for more descriptive statistics).

**Table 1. Descriptive statistics of participating and non-participating students.**

| | Non-participants | | Participants | |
|---|---|---|---|---|
| | N | %/mean | N | %/mean |
| Gender | | | | |
| *Girls* | 27,264 | 48.92% | 5,481 | 51.21% |
| *Boys* | 28,471 | 51.08% | 5,223 | 48.79% |
| Migration background | | | | |
| *Dutch background* | 39,770 | 71.36% | 6,683 | 62.43% |
| *Western background* | 11,202 | 20.10% | 3,069 | 28.67% |
| *Non-western background* | 4,763 | 8.55% | 952 | 8.89% |
| Parental education level | | | | |
| *Low educated* | 4,90 | 8.78% | 1,610 | 15.04% |
| *Average educated* | 16,893 | 30.11% | 3,894 | 36.38% |
| *High educated* | 29,058 | 52.14% | 4,033 | 37.68% |
| *Unknown* | 4,884 | 8.76% | 1,167 | 10.90% |
| Parental income level | | | | |
| *Low income* | 11,479 | 20.60% | 3,221 | 30.09% |
| *Average income* | 29,866 | 53.59% | 5,551 | 51.86% |
| *High income* | 14,160 | 25.41% | 1,873 | 17.50% |
| Household structure | | | | |
| *Two-parent family* | 46,138 | 82.78% | 8,420 | 78.66% |
| *One-parent family* | 9,597 | 17.22% | 2,284 | 21.34% |
| Parental labor market position | | | | |
| *Both parents work* | 41,067 | 77.91% | 6,762 | 68.75% |
| *Only father works* | 6,897 | 13.09% | 1,647 | 16.75% |
| *Only mother works* | 2,219 | 4.21% | 521 | 5.30% |
| *Both parents don't work* | 2,374 | 4.50% | 864 | 8.78% |
| Grade in primary education | | | | |
| *Grade 2* | 14,888 | 26.71% | 2,988 | 27.91% |
| *Grade 3* | 16,214 | 29.09% | 3,048 | 28.48% |
| *Grade 4* | 16,346 | 29.33% | 3,083 | 28.80% |
| *Grade 5* | 8,287 | 14.87% | 1,585 | 14.81% |
| End-of-year test $t_0$ | | | | |
| *Composite score* | 22,861 | 0.059 | 4,360 | -0.486 |
| *Reading* | 22,861 | 0.049 | 4,360 | -0.482 |
| *Mathematics* | 22,861 | 0.068 | 4,360 | -0.491 |
| End-of-year test $t_1$ | | | | |
| *Composite score* | 32,874 | 0.069 | 6,344 | -0.421 |
| *Reading* | 32,874 | 0.064 | 6,344 | -0.417 |
| *Mathematics* | 32,874 | 0.075 | 6,344 | -0.425 |

## Test scores

Students in Dutch primary education take standardized tests twice a year; the midyear test usually takes place in January or February, and the end-of-year test takes place in May or June. This paper analyzes the test scores for reading and mathematics from the end-of-year test in 2019/2020 ($t_0$) and the end-of-year test in 2020/2021 ($t_1$). As the remediation programs took place in the school year 2020/2021, the $t_0$ test is taken right before the programs started, and the

$t_1$ test is taken around the official end date of the programs. All test scores are standardized by grade level within each domain to enable comparison across domains and interpret the effects. This standardization is necessary as test scores from different domains are not directly comparable. Within a domain, however, the test scores are comparable, as they are measured on a continuous scale. Since the progression of test scores varies between grade levels, test scores are standardized by grade level as well. Table 1 shows descriptive statistics for the test scores of participating and non-participating students before ($t_0$) and after the program ($t_1$), more information can be found in S1 and S3 Appendix.

## Characteristics of remediation programs

Additionally, we have information about the characteristics of the remediation program. Schools were required to complete a questionnaire detailing their proposed remediation program to secure the funds for the programs. Based on this questionnaire, we received information about the characteristics of the remediation programs (see S1 Appendix for more information regarding the questionnaire). We use the following information regarding the remediation programs: the organization, the timing, the group size, the goals, and the type of support the programs offer. As for organization, schools can choose to organize it by themselves or collaborate with external parties, leading to a difference in whether internal or external staff are deployed. The program can take place during or outside regular school hours (regular hours in primary education are typically between 8:30 am and 3 pm). Furthermore, support is offered to students of different group sizes. The program goals are math or reading (or both), or non-cognitive skills such as study and socio-emotional skills. Lastly, the type of support includes additional support and guidance, purchase of new methods, extended instructions, extended school days, remedial teaching, support during work, or unknown/else. We have created five variables with mutually exclusive categories within each variable regarding the characteristics of remediation programs, which means that for each variable, every school appears in one category of the response options (see S3 Appendix for more information about the operationalization of these characteristics). Table 2 shows the descriptive statistics of the characteristics of the remediation programs for all participating students (N = 10,704). The remaining students, the non-participating students (N = 55,735), are our control group and therefore do not have these program characteristics.

## Analyses

The analyses of this paper consist of three parts [47]. We estimate equation 1 with the logistic package and equations 2 and 3 with the xtreg package in Stata 15. First, we assess the likelihood of participating in remediation programs $y_i$ with a logistic regression with school-level clustered standard errors, based upon student background characteristics $X_i$ and previous test scores $A_i$, where $\mu_i$ represents unobserved characteristics influencing the selection of students into remediation programs and $\varepsilon_{is}$, is the error term.

$$y_i = \beta_0 + \beta_1 X_i + \beta_2 A_i + \mu_i + \varepsilon_{is} \tag{1}$$

To assess the impact of participating in a remediation program on students' achievements, we employ a Difference-in-Differences analysis with an ordinary least squares regression with school-level clustered standard errors and school-level fixed effects. We distinguish between participants ($p_1$) and non-participants ($p_0$) and a time indicator where $t_0$ captures the tests taken before the start of remediation programs and $t_1$ at the end of school year 2020/2021. Furthermore, vector $X_{it}$ consists of student background characteristics and $S_{it}$ of school-level information. Additionally, we include school fixed effects $\mu_i$ and $\varepsilon_{ii}$ as error term clustered at the school level. The DiD-estimator of interest is $\beta_3$.

$$y_{it} = \beta_0 + \beta_1 p_i + \beta_2 t_{it} + \beta_3 p_i t_{it} + \beta_4 X_{it} + \beta_5 S_{it} + \mu_i + \varepsilon_{is} \tag{2}$$

**Table 2. Descriptive statistics of the characteristics of remediation programs.**

|  | % of participating students |
|---|---|
| Organization of remediation program |  |
| *Internal staff* | 15.64% |
| *External staff* | 2.10% |
| *Internal & external staff* | 18.41% |
| *Unknown* | 0.81% |
| *Students without information* | 63.03% |
| Moment of remediation program |  |
| *Program outside regular hours* | 4.10% |
| *Program during regular hours* | 4.83% |
| *Program outside & during* | 10.35% |
| *Unknown* | 17.68% |
| *Students without information* | 63.03% |
| Group size |  |
| *Entire class* | 1.37% |
| *Small groups (2–5 students)* | 2.43% |
| *Small groups (6–10 students)* | 0.64% |
| *Groups of unknown size* | 7.99% |
| *Individual* | 0.77% |
| *Individual and groups* | 4.24% |
| *Unknown* | 19.52% |
| *Students without information* | 63.03% |
| Goal |  |
| *Language and math goal* | 15.73% |
| *Language goal* | 3.20% |
| *Mathematic goal* | 1.60% |
| *Cognitive and non-cognitive goal* | 14.87% |
| *Non-cognitive goal* | 1.58% |
| *Students without information* | 63.03% |
| Type of support offered |  |
| *Additional support and guidance* | 11.01% |
| *Purchase of new methods* | 5.35% |
| *Extended instructions* | 3.92% |
| *Extended schoolday* | 2.73% |
| *Remedial teaching* | 0.79% |
| *Support during work* | 0.76% |
| *Unknown* | 12.37% |
| *Students without information* | 63.03% |

Lastly, we analyze whether the characteristics of remediation programs affect students' achievements with a Difference-in-Differences analysis. These analyses are conducted by incorporating an interaction effect of the participation treatment group and interventions in the remediation programs ($R_i$), where we distinguish between students who have not participated in a program ($R_0$), students who have participated in a program with characteristic $R$ ($R_1$) and students who have participated in a program with unknown characteristic ($R_2$). We include school fixed effects $\mu_i$ and error term $\varepsilon_{il}$ clustered at the school level. Moreover, we include interaction effects to control for the effect of all other characteristics regarding the remediation programs ($R_{other}$). The DiD-estimator of interest is $\beta_3$.

$$y_{it} = \beta_0 + \beta_1 R_i + \beta_2 t_i + \beta_3 t_i R_i + \beta_4 t_i R_{other_i} + \beta_5 X_{it} + \beta_6 S_{it} + \mu_i + \varepsilon_{is} \tag{3}$$

## Results

### Participation in remediation programs

The remediation programs were explicitly set up for the most disadvantaged and vulnerable students affected by the COVID-19-induced school closures. As Fig 1 shows, selection into the programs was not random. We find that participation is linked to indicators of disadvantage. The results indicate that students with less educated parents are more likely to participate in remediation programs (1.5% more likely than the mean), whereas students with both parents in employment are less likely to participate (1% less likely than the mean). There is also a strong inverse relationship between test scores and participation in remediation programs. Students with extremely low test scores (-6 SD) have the highest probability of participation (90%), while those with average (0 SD, 15%) or above average scores (3 SD, 2%; 4 SD, 1%) are considerably less likely to participate. This pattern suggests that students with lower test scores are more inclined to participate, with the probability decreasing sharply as scores increase. This implies that lower educational achievement is an important indicator for participation.

### Average effect of remediation programs

To assess whether participation in remediation programs leads to an increase in students' achievements, we employ a Difference-in-Differences (DiD) analysis [48]. By utilizing this method, we explicitly include all students' achievements before ($t_0$) and after participation ($t_1$) in remediation programs. The crucial assumption of the DiD analysis is the parallel time trend assumption [48]. Fig 2 displays the time trends for the cohorts of students we include in our analyses. The results in Fig 1 have already shown that selection into the treatment is not random, rather that lower educational achievement is an important indicator for participation. In the pre-trends in Fig 2, we see divergence in progress between students who participated in the programs and non-participating students, reflecting the obvious differences in early learning losses after the first school closures, making it important to consider that the overall effects of the program might be biased in either direction. Interpreting the results in this paper as a causal effect of the remediation programs requires caution. However, this approach aims to approximate causality, considering that these programs have a non-random treatment group.

To estimate the DiD analyses, the test scores are standardized and pooled into an average composite score encompassing both reading and mathematics test scores. Table 3 presents the results of the DiD for this composite score and both domains separately (see S5-S7 Tables). The models include the DiD-estimator (the post-treatment indicator for participating students), which indicates the average growth in test scores after participation in the remediation programs. The DiD-estimator shows a significant and positive interaction effect, suggesting that students who participated in remediation programs scored an additional 0.05 SD (composite score) on their tests at time $t_1$. Nevertheless, we find that participating students still have significantly lower test scores on average (-0.54 SD). Furthermore, separate analyses per grade were conducted, as Bloom et al. [49] show that growth during the school career is not linear, the results show that students in the higher grades show the largest increases in achievements after participation (see S8 Table).

### Characteristics of remediation programs

Lastly, we examine which remediation programs have the strongest relationship with learning growth, as these programs show a rich diversity in approaches. We investigate whether differences in the organization, timing, group size, goals, and type of support of the remediation programs affect the average growth in test scores following participation in the remediation programs. Fig 3 shows the results of these five aspects of remediation programs. We find that the timing and organization of remediation programs are not related to the learning growth of participants in remediation programs. The group size, program goal(s) and type of support show significant relationships with learning growth.

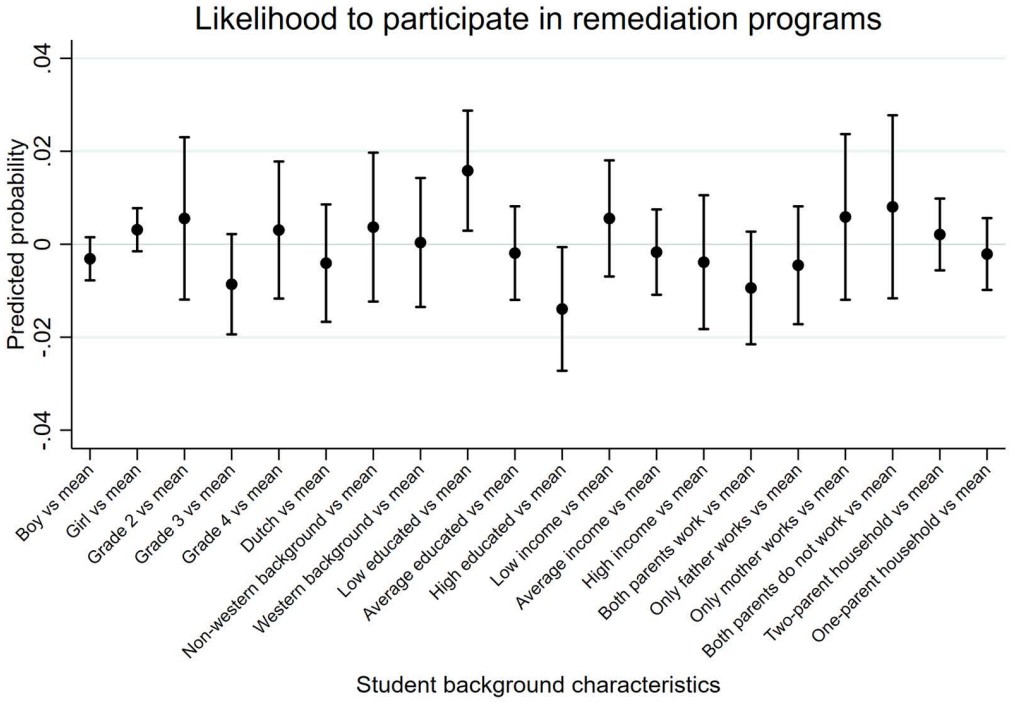

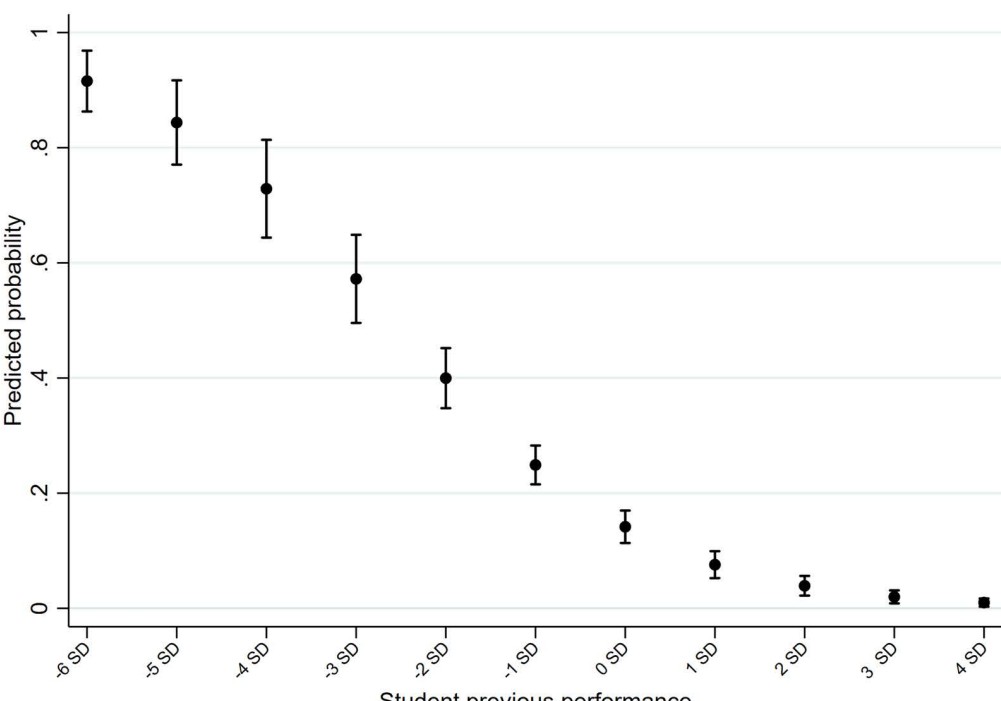

**Fig 1. Likelihood of participating in remediation programs based on student background and previous performance.** *The fig reports predicted probabilities based upon logistic regressions with school-level clustered errors. Previous performance is measured as the learning loss between the end-of-year test and the midyear test in school year 2019/2020 and consists of a composite score encompassing mathematics and reading. The figs show 95% confidence intervals. See S1-S4 Tables for the underlying regression models and robustness checks.*

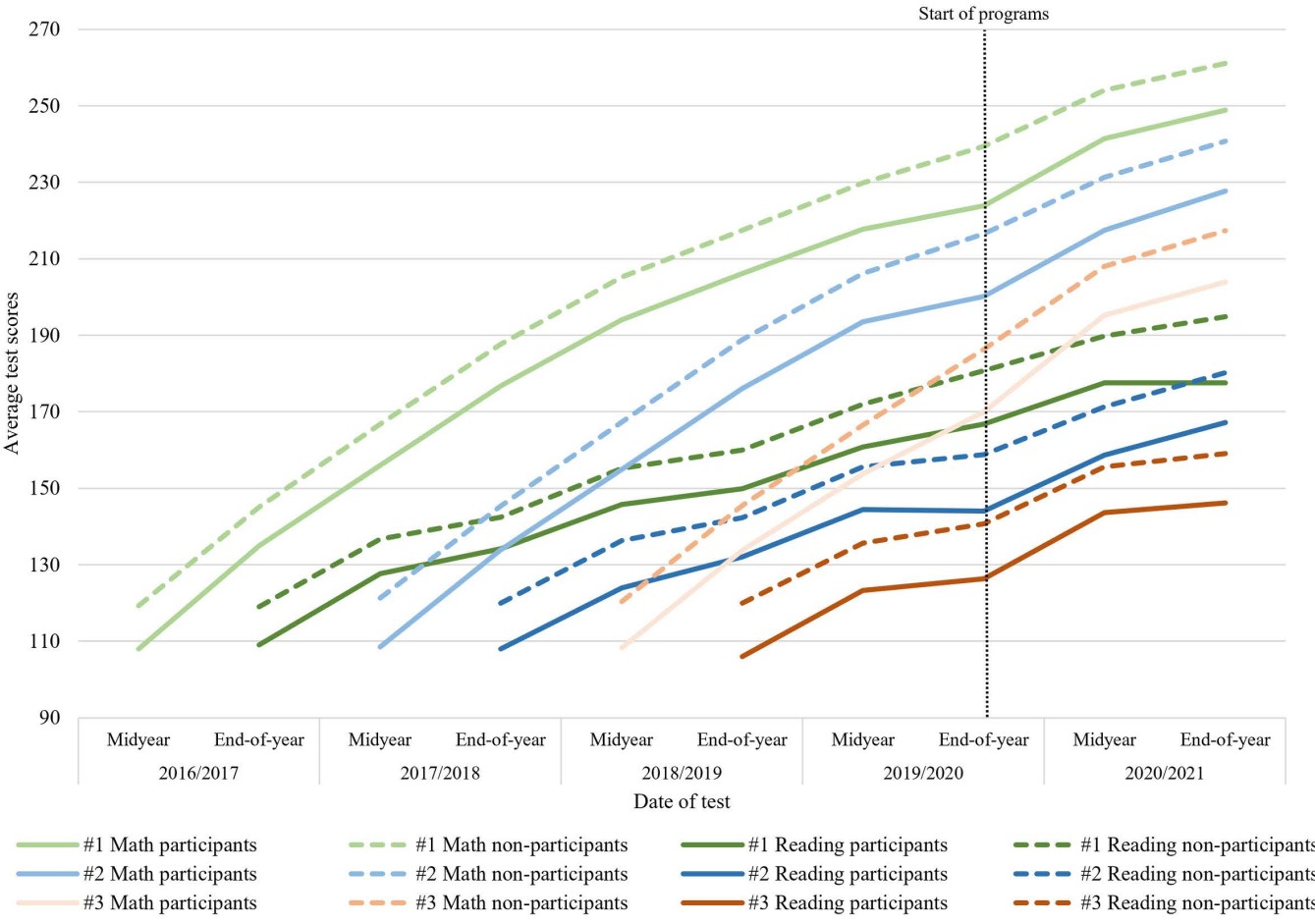

**Fig 2. Development of test-scores for the different cohorts in the sample.** *This fig shows the parallel time trends for participants and non-participants in mathematics and reading for the three cohorts used in our analyses both before and after the remediation programs. The fig shows the pre-trends up until the end-of-year test in 2019/2020 and the post-trends in school year 2020/2021. The vertical line in the fig represents the transition from pre-trends to post-trends. The first cohort (green) started grade 1 of primary education in 2016/2017. The second cohort (blue) began in 2017/2018, and the third cohort (orange) started in 2018/2019. As reading does not have a midyear test in grade 1, the trends for reading start with the end-of-year test in grade 1.*

The results show that students in individual remediation programs have significant positive increases in their test scores for reading comprehension compared to non-participants. For mathematics, on the other hand, programs in small groups, approximately between 2 and 5 students, have positive effects on test scores. For the composite score, we find positive effects for programs in both small groups and individually.

In general, programs that have set goals to improve the cognitive skills of their students show significant positive results compared to non-participants. Some programs aim to improve language and mathematic skills, whereas others focus on either. For students' composite scores, we see that programs with a combination of language and mathematical goals also show positive significant effects. For programs that have only set a goal for mathematical skills, we find positive and significant effects on both students' language and mathematics test scores. Programs with non-cognitive goals or a combination of cognitive and non-cognitive goals show no significant effects on students' test scores. This can be attributed to the fact that these programs concentrated on enhancing study skills or socio-emotional skills. Although these programs did not result in an immediate increase in test scores, it is plausible that they may contribute to long-term academic success, by improving learning skills.

**Table 3. Difference-in-Differences analysis for participants and non-participants.**

| | Composite | Reading | Mathematics |
|---|---|---|---|
| School year 2020/2021[a] | 0.010 | 0.015^ | 0.006 |
| | (0.007) | (0.008) | (0.008) |
| Participating students[b] | -0.537*** | -0.514*** | -0.560*** |
| | (0.014) | (0.016) | (0.016) |
| School year * Participation | 0.052** | 0.050** | 0.054** |
| | (0.017) | (0.019) | (0.019) |
| Student controls | Yes | Yes | Yes |
| School level controls | Yes | Yes | Yes |
| School-level fixed effects | Yes | Yes | Yes |
| Constant | -0.242* | -0.447** | -0.036 |
| | (0.123) | (0.140) | (0.140) |
| Observations | 66,439 | 66,439 | 66,439 |
| R-squared | 0.138 | 0.121 | 0.140 |
| Clusters | 456 | 456 | 456 |

Note: Robust standard errors in parentheses; *** $p < 0.001$, ** $p < 0.01$, * $p < 0.05$, ^ $p < 0.1$. [a] the reference category is the school year 2019/2020; [b] the reference category is students who did not participate in the remediation programs but are enrolled in schools that offer remediation programs. Student controls include sex, migration background, parental education and income, and household structure, and school-level controls include denomination, urbanization, and the disadvantage score of the school.

Finally, the type of support offered to students in the programs is examined. Here, we do not find many significant results, suggesting that the specific type of support provided may be less important than simply ensuring that students receive additional support. However, for reading, we identify two significant negative relationships. First, students in programs offering extra support (without a clear specification of its nature) score significantly lower compared to students not in a remediation program, possibly because these schools and programs lacked a structured action plan. Second, students in programs that extended the school day also showed significantly lower reading test scores compared to non-participants. This finding aligns with existing literature suggesting that longer teaching hours can sometimes have unintended negative effects.

**Robustness checks**

We performed several robustness checks. First, we used Inverse Probability Weights (IPW-weights) to align our sample with all students in Dutch primary education. Our sample over-represents vulnerable and disadvantaged students in Dutch primary education, including more non-western migrants, low-educated parents, low-income, and single-parent families than the national average. The schools are in more urbanized areas with higher disadvantage scores. This is expected, as remediation programs focus on these students. Hence, we also estimate our models where we include the IPW-weights estimated based upon the entire population of students. These results confirm our main findings (see S9 Table).

Second, we have conducted our main estimation model again for only a selection of the sample. We divided our sample in half based on student's performance before the remediation programs (as it was specifically the lower-performing half of the students that was targeted with the remediation programs). The higher-performing students might bias the estimates and influence the pre-trends of the control group (See S1 Fig for the parallel time trends). The results of limiting our sample to lower-performing students are consistent with the main finding for composite and reading. However, for mathematics, the effect is not significant anymore in this robustness check. Therefore, these results need to be treated with more caution (see S10 Table).

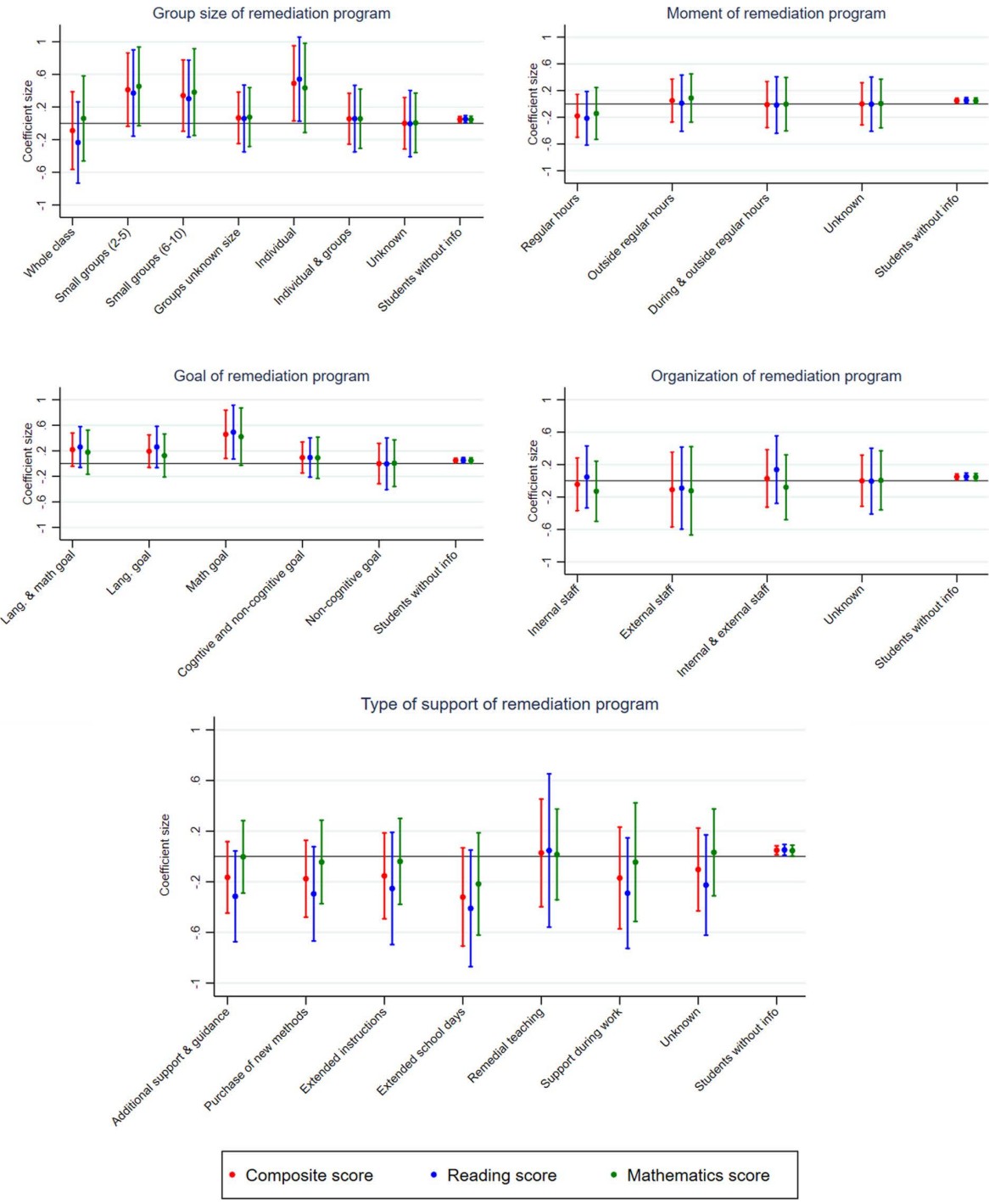

**Fig 3. Characteristics of remediation programs on students' achievements (DiD estimator).** *The fig reports the coefficients of the Difference-in-Differences estimator based on five regression analyses on the characteristics of remediation programs. The figs show 95% confidence intervals. See S11-S15 Tables for detailed results and the underlying regression models.*

Lastly, we estimated our main models using Propensity Score Matching (PSM) approaches to align our control and treatment groups. Using the Kernel matching procedure allows us to weigh observations depending on their proximity to the treatment group based on background characteristics and previous test scores, with the main advantage that we hardly lose any students in our sample (more information about the PSM approach can be found in S4 Appendix). This approach was used to align our treatment and control group better and create more similar pre-trends (see S2 Fig for the parallel time trend). We continue to use our DiD approach to analyze the effects of the remediation programs and include the PSM weights to better align our sample. The overall results confirm our main findings (see S4 Appendix).

## Conclusion and discussion

COVID-19 has created an enormous crisis in education and learning [2], with vulnerable and disadvantaged students in particular facing large learning losses due to school closures [4–7,10,12,13]. Around 40% of all countries made various efforts at remediation. Here, we study the effectiveness of one remarkably swift and ambitious attempt at remediation. Upon recognizing the learning losses experienced by students during the COVID-19 pandemic, the Dutch government allocated funds to schools, allowing them to establish remediation programs based on their own preferences. We show that vulnerable students with disadvantaged backgrounds were selected for the remediation programs (research aim 1), and that there was a significant positive increase in test scores after participation of 0.05 SD (research aim 2). Especially individual or small-scale remediation programs, and programs with a sole emphasis on cognitive skill development, result in an increase in achievements after participation (research aim 3).

Our study shows that remediation programs that address content gaps due to the COVID-19 pandemic can be effective. This approach is not new; however, the rapid implementation of these programs and the diversity of schools have resulted in varying levels of success. Although not universally effective, many programs have shown enough promise to justify allocating funds for their development. Furthermore, the results show that schools can manage this autonomy in the Dutch context, where schools enjoy considerable autonomy. Our study shows that the students who most needed support were correctly identified and enrolled in these remediation programs. Furthermore, schools were able to develop and implement programs tailored to the needs of these students. This suggests a robust capacity within schools to use their autonomy in ways that benefit students, particularly in challenging circumstances such as those posed by the pandemic.

Our Difference-in-Differences (DiD) analyses estimate the effect of remediation programs at approximately 0.05 SD, with a 95% confidence interval ranging from 0.019 to 0.085 SD. While this effect size may appear modest [50], it signals a meaningful and positive impact. Given the relatively short duration of these programs, typically between 25 and 40 hours, this gain is notable. To put this into perspective, the average yearly learning progress in primary education is about 0.4 SD [51] under normal circumstances, a benchmark that was likely unattainable during the pandemic. Even within these constraints, remediation programs contributed to students' progress. Considering the time investment, the observed effect suggests that remediation programs provided a positive return in terms of learning gains relative to the limited instructional hours. Therefore, these remediation programs are a solution to the COVID-19 related learning losses. Relative to the 0.08 SD learning loss reported by Engzell et al. [5] or the 0.15 SD learning loss reported by Haelermans et al. [7], the remediation programs have successfully mitigated between half to one-third of the learning losses obtained in the first half year of the pandemic.

Given the divergence in the pre-trends regarding test score progress between the midyear test of 2019/2020 and end-of-year test in 2019/2020 among participating and non-participating students, it is important to consider that the overall effects of the program might be biased in either direction. The estimate may be biased upward if the participating students would have recovered without the remediation programs, such as due to the reopening of schools in general. Conversely, it may be biased downward if disadvantaged students, who were more likely to participate, would have continued to experience lower learning growth without the support of remediation programs. Interpreting the results in this paper as a causal effect of the remediation programs requires caution. However, this approach aims to approximate causality as

best as possible, considering that these programs have a non-random treatment group. While it remains possible that our estimates are subject to both upward and downward bias, we have taken the necessary steps to strengthen their robustness. Overall, the evidence indicates that remediation programs were effective, though the precise magnitude of the effect may vary depending on the chosen specification. The same caution is needed when interpreting the estimates regarding the different characteristics of remediation programs as these estimates may be subject to bias or may be confounded by unobserved heterogeneity. Our robustness checks, including PSM models and analyses focusing on the lower-performing half of students, show somewhat more comparable pre-trends. The results of these robustness checks are largely consistent with our main outcomes.

It is important to note that while the effect sizes of interventions mentioned in previous studies in the main text were often more substantial, these were frequently measured in controlled experimental settings. Proven effective interventions in the past do not necessarily lead to effective interventions in the future [52], as interventions are seldom carried out as designed, partly explaining differences in expected outcomes. Furthermore, these controlled interventions were often conducted in small-scale settings with close involvement from developers, leading to high-quality implementations that may not be replicable in larger, real-world settings [33]. The fact that our research identifies significant effects in a natural context underscores the relevance of these interventions, even in less-than-ideal conditions. Additionally, we must consider that the remediation programs have been designed and implemented in a turbulent school year. Besides the chaotic time during the COVID-19 pandemic, a nationwide teacher shortage made designing and implementing the programs difficult. Lastly, a second school closure might have altered the planning and timing of the remediation efforts. This second school closure in the Netherlands took place from December 16th, to February 8th, leaving schools closed for another 7.5 weeks (including two weeks of Christmas holidays).

This research is among the first to assess remediation programs after the COVID-19 pandemic at the national level. The diversity of remediation programs offered in the Netherlands allows us to study the effectiveness in a comparative setting. Whereas much previously mentioned research focuses on reading or mathematics, our research adds to existing knowledge by analyzing all programs on reading and mathematics. Besides, previous research into remediation programs has been (mostly) conducted outside of the Netherlands in different educational settings, mainly focused in the US and UK, for example, research by Slavin [27] and the Education Endowment Foundation [20–26].

Previous research showed that remediation efforts faded quickly [53]. Future research could track students over a more extended period and look at the long-term outcomes of these programs on vulnerable students, specifically the likelihood of grade retention in the upcoming years. The hope is that with remediation, students will be able to keep up better in their regular classes and not have to repeat a year. The Netherlands has since funded a National Program for Education, called the NPO, with which remediation initiatives are extended up to 2023. Future research could also look at the effects of a lengthier funding initiative over the past years since the COVID-19 pandemic.

These results were found in one country that was unusually fast in setting up funds for remediation programs. It is an open question to what extent results would replicate elsewhere. One factor that would support generalization is that individual schools developed programs, which resulted in a wide array of program goals, designs, and implementations, unlike where one actor was chosen for design and implementation. The results found in this study may thus represent an average of what different centrally designed, uniformly implemented programs (such as that developed by EEF for schools in England and Wales [15]) would reach.

## Supporting information

**S1 Appendix. Data collection.**
(PDF)

**S2 Appendix. Representativeness of the sample.**
(PDF)

**S3 Appendix. Descriptive statistics and operationalization.**
(PDF)

**S4 Appendix. PSM-matching approach.**
(PDF)

**S1 Table. The likelihood of participating in remediation programs (learning loss indicator).**
(PDF)

**S2 Table. The likelihood of participating in remediation programs (end-of-the-year test scores).**
(PDF)

**S3 Table. The likelihood of participating in remediation programs for the remediation programs sample.**
(PDF)

**S4 Table. The likelihood of participating in remediation programs for the cognitive remediation programs sample.**
(PDF)

**S5 Table. Students' achievements before and after remediation programs – Composite score.**
(PDF)

**S6 Table. Students' achievements before and after remediation programs – Reading score.**
(PDF)

**S7 Table. Students' achievements before and after remediation programs – Mathematics score.**
(PDF)

**S8 Table. DiD-estimator for analyses separately per grade on students' achievements before and after remediation programs.**
(PDF)

**S9 Table. Students' achievements before and after remediation programs with IPW-weights.**
(PDF)

**S10 Table. Student's achievements before and after remediation programs with lower performing students only (half of sample).**
(PDF)

**S11 Table. Effect of remediation program's group size on students' achievements.**
(PDF)

**S12 Table. Effect of moment of the remediation program on students' achievements.**
(PDF)

**S13 Table. Effect of remediation program's organization on students' achievements.**
(PDF)

**S14 Table. Effect of remediation program's goal on students' achievements.**
(PDF)

**S15 Table. Effect of remediation program's support on students' achievements.**
(PDF)

**S1 Fig. Development in test scores cohort #1, #2 and #3 – lower performing half of sample.**
(PNG)

**S2 Fig. Development in test scores cohort #1, #2 and #3 – PSM sample (Kernel bandwidth 0.06).**
(PNG)

## Acknowledgments

The authors thank two anonymous reviewers, Bart Golsteyn, Babs Jacobs, Kars van Oosterhout, Nico Pestel, and Sanne van Wetten for their helpful comments on previous versions of this paper.

## Author contributions

**Conceptualization:** Madelon Jacobs, Carla Haelermans, Martijn Meeter.

**Data curation:** Madelon Jacobs, Martijn Meeter.

**Formal analysis:** Madelon Jacobs.

**Funding acquisition:** Carla Haelermans, Martijn Meeter.

**Methodology:** Madelon Jacobs, Carla Haelermans, Martijn Meeter.

**Project administration:** Carla Haelermans, Martijn Meeter.

**Software:** Madelon Jacobs.

**Supervision:** Carla Haelermans, Martijn Meeter.

**Visualization:** Madelon Jacobs.

**Writing – original draft:** Madelon Jacobs.

**Writing – review & editing:** Madelon Jacobs, Carla Haelermans, Martijn Meeter.

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
