## [Decision Letter · Decision Letter 0]

19 Aug 2024

PONE-D-24-17313Effective Remediation Programs for Vulnerable Students to Overcome Learning LossPLOS ONE

Dear Dr. Jacobs,

Thank you for submitting your manuscript to PLOS ONE. After careful consideration, we feel that it has merit but does not fully meet PLOS ONE’s publication criteria as it currently stands. Therefore, we invite you to submit a revised version of the manuscript that addresses the points raised during the review process. Both reviewers recommended major revisions to the manuscript and I agree with this assessment, while also being confident that this paper certainly has the potential to be suitable for publication with appropriate revisions. As such, please do read and respond to both sets of reviewers' comments. Reviewer 1 provides various useful suggestions that I would encourage you to consider carefully and respond to in your revision, as I agree with their point that these could help to enrich the paper in places. In terms of changes required for acceptance, however, I would especially draw your attention to a number of very helpful comments and suggestions provided by Reviewer 2 in ensuring your conclusions are appropriately supported by your analyses. I'd summarise the most pressing issues to support acceptance as follows:

I have some concerns in relation to your current data availability statement (also noted by Reviewer 2 in their selection of the 'No' option in relation to the data availability review question) and its compatibility with the journal's stated policy in this area. While I am sympathetic to the fact that not all data will be shareable due to legal or ethical concerns, PLOS' policy requires you to be considerably more specific with the information you provide on with whom readers can make contact about the possibility of accessing the data. For example, the statement "Our access was granted through a partnership with a non-profit organization that established special agreements to facilitate this research. Researchers seeking access to the same data must engage in a similar partnership arrangement." needs to be amended to provide the details required by the data availability policy section on acceptable restrictions in relation to third party data on this page: https://journals.plos.org/plosone/s/data-availabilityConcerns regarding the analytical strategy for research aim 3, noted by Reviewer 2, are something I'd definitely want to see addressed as this appears likely to improve the clarity and interpretation of the results here.I would also strongly encourage you to make revisions in response to Reviewer 2's concerns regarding pre-trends and the possibility that you may be estimating a lower bound of the actual effect. This doesn't risk undermining your conclusions, but it's important to note.Both reviewers suggest the potential the strengthen the conceptual discussion of programme characteristics and the potential benefit of using a theoretical framework to discuss their importance.

We look forward to receiving your revised manuscript.

Kind regards,

Jake Anders

Academic Editor

PLOS ONE

Journal Requirements:

"This research was supported by the Netherlands Initiative for Education Research (NRO), project number 40.5.20937.001. "

Reviewers' comments:

Reviewer's Responses to Questions

**Comments to the Author**

1. Is the manuscript technically sound, and do the data support the conclusions?

Reviewer #1: Yes

Reviewer #2: Partly

2. Has the statistical analysis been performed appropriately and rigorously? 

Reviewer #1: Yes

Reviewer #2: Yes

3. Have the authors made all data underlying the findings in their manuscript fully available?

Reviewer #1: Yes

Reviewer #2: No

4. Is the manuscript presented in an intelligible fashion and written in standard English?

Reviewer #1: Yes

Reviewer #2: Yes

5. Review Comments to the Author

Reviewer #1: Thank you for such a nice research. I have a few recommendations to enrich the paper, as following:

1. Research Question: Ensure that the research question is explicitly stated.

2. Sample Size: On page 6, correct the sample size to 66,500 (not 66.500).

3. Literature Review Citations: In the introductory paragraph of the literature review on page 6, support claims about existing research and numerous studies with proper citations in the main text and reference section.

4. Intervention Examples: In the second paragraph of the literature review on page 6, list specific examples of various interventions. Clarify which interventions, such as one-on-one tutoring, are most effective.

5. Areas of Target in Intervention: Explicitly describe the particular areas targeted by the intervention.

6. Citations for Previous Research: The second sentence of the last paragraph on page 8 needs citations for statements related to previous research. Include corresponding references in the reference section.

7. Thematic Review of Learning Loss and Remediation: Enrich the literature review by analyzing learning loss thematically and discussing specific themes related to remediation.

8. Method Section Variables: Enhance the method section by explicitly describing variables such as participation in the remediation program, impact of the program, and effective remediation strategies. Provide a theoretical framework for clarity.

9. Data Analysis Tools: Specify the tools used for data analysis in the method section of the main text.

10. Policy and Pedagogical Implications: Conclude the discussion section by adding policy and pedagogical implications based on the study’s findings

Reviewer #2: This manuscript aims to (1) show which social groups participated in the national remedial education program in the Netherlands during the first year of the COVID-19 pandemic, (2) estimate the overall effect of the national remedial education program on children’s test performance in math and reading, and (3) examine how program effects co-vary with the characteristics of specific remedial education measures chosen by different schools.

The topic of the manuscript and the three specific research aims are of clear importance and worthy of investigation. The main conclusions of the manuscript are supported by the data analysis. That said, I would suggest some adjustments in the interpretation of the results concerning research aim (2) and in the analysis and interpretation of results concerning research aim (3). I will explain these below and hope that they will be of help to the authors in further strengthening this important piece of research. I will also offer some minor suggestions for further improvements.

-> Analyses and interpretation of research aim (2): To estimate the overall effect of the national remedial education program on children’s test performance in math and reading, the manuscript compares the performance gains of students who participated in the program and those who did not, between June 2020 and June 2021. Given that students who participated in the program had a lower baseline performance and were from more disadvantaged backgrounds, estimating overall effects in this difference-in-difference design requires that the learning rate of the two comparison groups is similar prior to the program. The manuscript acknowledges the importance of this parallel trends assumption and refers to section 7 of the Supplementary Information where this assumption is tested. While I agree that figures S7.1-4 suggest that learning progress is similar across the two comparison groups for most time periods, there is a notable divergence in the trends in reading progress between January/February 2020 and June 2020 for cohorts 1 and 2 (see F7.1-2 on p42-43 of the SI). Arguably, this time period is most relevant, as it is right before the remedial education program was implemented. This divergence in learning progress is not surprising, as it reflects early learning losses during the COVID-19 pandemic in the Netherlands (see Haelermans et al. 2022 and Engzell et al. 2021). These learning losses are known to be concentrated amongst the groups of students from more disadvantaged backgrounds, who also had a higher likelihood of participating in remedial education programs, as shown by the manuscript in the analyses for research aim (1). It would be important to take this divergence into account when interpreting the results for research aim (2). The divergence in learning progress may suggest that the estimates of the overall effects of the remedial education program are lower-bound estimates, as it is likely that the learning rate of children with a lower baseline performance and from lower socioeconomic backgrounds would have continued to have a(n even) lower learning rate in the absence of the remedial education program.

-> Analyses and interpretation of research aim (3): As its third main aim, the manuscript seeks to examine how program effects co-vary with the characteristics of specific remedial education measures implemented in different schools. To this end, authors run a series of 13 models in which they successively replace the binary treatment/control dummy variable used for the analyses of research aim (2) with multi-category measures of different characteristics across which different remedial education measures vary and which are thought to be relevant for children’s learning progress. The results from these models are shown in Tables S6.1-13 in the SI. They then select three of these characteristics (remedial teaching, group size, and staff) and combine them in the coefficient plot shown in Figure 2.

It is not clear to me that the chosen analytical approach is best suited to achieve the third research aim of the study and I suggest an alternative approach further below. A key problem is that the characteristics of the remedial education measures implemented in different schools are likely to co-vary and be confounded by characteristics that are not included in each of the separate models shown in Tables S6.1-13. This may explain counterintuitive results such that remedial education measures with a focus on language have no positive effect on children’s reading progress, but do have an effect on children’s math progress (see Table S6.8 on p.35 of the SI). In short, not including different program characteristics in the same model considerably limits the extent to which the associations shown between effect sizes and program characteristics can be interpreted as causal relationships. It would be important to highlight this in the text.

A related problem of the current analytical approach is that one cannot gauge the statistical significance of the differences in the effect sizes between different remedial education measures. This is because students who participated in a given remedial education measure are compared to students who did not participate in any remedial education program, rather than students who participated in a remedial education measure that differed on the relevant characteristic in question. Relatedly, the current interpretation of some of results of Figure 2 reported on p.15-16 are not technically correct. While the interpretation concerning ‘remedial teaching’ is correct, the interpretation of the results concerning group size should be adjusted. Currently the manuscript states that “… positive outcomes were associated with small-group programs as opposed to larger settings, with a 0.26 SD higher increase in the test scores for participants with small-group remediation programs.” This is somewhat misleading, as it suggests that there is a substantively large and statistically significant difference in the effects of remedial education measures characterised by small groups, vs larger groups. Yet, as Figure 2 shows, the difference between ‘entire class’ and ‘small groups’ is substantively relatively small and (telling from the confidence intervals) not statistically significant. It would be important and relatively straightforward to adjust the interpretation to reflect the fact that the reference group here are students who did not participate in remedial education and there seems to be no statistically significant difference in the effectiveness of remedial education measures with different group sizes.

An alternative analytical approach to address research aim (3) would be to extract effect sizes of the remedial education program for each school in the sample and use these as the dependent variable with program characteristics as the focal independent variables, controlling for school characteristics and characteristics of the student body participating in the remedial education program. Effect sizes could be weighed according to the precision of each estimate to account for cross-school variation herein. An important advantage of this alternative analytical approach would be that one could include all relevant program characteristics in the same model and thereby avoid the confounding between program characteristics that likely limits the interpretability of the results in the current version of the manuscript. The proposed alternative approach would also show the substantive size and statistical significance of the association between different program characteristics and children’s learning progress. It would also facilitate jointly showing and discussing the coefficients of all program characteristics analysed rather than focussing on only three characteristics in the main text, as is currently done.

It would be helpful to explain more clearly what is meant by the concepts used to describe the different program characteristics (e.g. what is meant by ‘remedial teaching’), as well as how they are operationalised and measured. Relatedly, it would be helpful to add more theoretical discussion of why the characteristics of different remedial education measures that are examined are thought to be relevant.

Minor points:

* The confidence intervals shown in Figure 1 and Figure 2 seem to reflect 0.10 thresholds for statistical significance, rather than the usual 0.05 thresholds (see, e.g., Tale S6.1). Given that this is relatively unconventional, it would be important to highlight this in the main text and in the description of the figures.

* It would be helpful to show results in Figure 2 for math and reading separately, given that effects vary by subject domain (as shown in Tables S6.1-13).

* It is not clear why the model with school-level fixed effects shown in Table S5.5 is not used as the preferred model in the main text for research aim (2). This would seem to be the most robust specification.

* The notes under Figure 1 state that “Students’ achievement is measured by the combined score of reading and mathematics on the difference between the midterm test and the end-of-year test in the school year 2019/2020.” This is not very clear and it would be helpful if it could be made clearer in the figure legend and main text what is meant here.

* It would be interesting and aid the interpretability of results to show descriptive statistics on the prominence of different characteristics of the remedial education measures chosen by different schools in the Netherlands. For instance, how many schools employed small group measures as opposed to measures with larger groups of students?

* It would be helpful to note that the analyses in Table S5.6 suggest that there are no differences in the effectiveness of the remedial education program across age groups. This could also be tested directly, using the alternative analytical approach outlined above.

* It would be helpful to add relevant references to the discussion of the existing evidence on remedial education programs on p.6-8. For instance what evidence does the reported effect size of 0.4 SD for one-to-one tutoring on p.7 (top) refer to?

* It would be helpful to state more clearly the three main aims/research questions in the introduction of the manuscript.

* Using predicted probabilities rather than odds ratios may aide the interpretability of Figure 1.

References:

Engzell P, Frey A, Verhagen MD. Learning loss due to school closures during the COVID-19 pandemic. Proceedings of the National Academy of Sciences. 2021;118(17).

Haelermans C, Jacobs M, van Vugt L, Aarts B, Abbink H, Smeets C, et al. A full year COVID-19 crisis with interrupted learning and two school closures: The effects on learning growth and inequality in primary education. ROA Research Memoranda No. 009

6. PLOS authors have the option to publish the peer review history of their article (what does this mean? ). If published, this will include your full peer review and any attached files.

**Do you want your identity to be public for this peer review?** For information about this choice, including consent withdrawal, please see our Privacy Policy .

Reviewer #1: **Yes: ** Shashidhar Belbase

Reviewer #2: No

---

## [Author Response · Author response to Decision Letter 1]

31 Oct 2024

Effective Remediation Programs for Vulnerable Students to Overcome Learning Loss

PLOSONE

October 2024

Thank you very much for all your valuable feedback and suggestions. Thanks to all the detailed comments and constructive criticism, we were able to significantly improve our paper. The insights have helped us to improve the quality and clarity of our work. We have carefully considered each of the comments and incorporated them into the revision of the paper. In the document below, we specifically address all points raised by both reviewers and the editor. Thank you for your time and effort.

Response to main points mentioned by academic editor

Thank you for providing us with the opportunity to revise the paper. We want to specifically address points 1 and 4 you raised, related to the data availability statement and the literature section of the paper. Points 2 and 3 will be addressed in the response to reviewers 1 and 2.

1. I have some concerns in relation to your current data availability statement (also noted by Reviewer 2 in their selection of the 'No' option in relation to the data availability review question) and its compatibility with the journal's stated policy in this area. While I am sympathetic to the fact that not all data will be shareable due to legal or ethical concerns, PLOS' policy requires you to be considerably more specific with the information you provide on with whom readers can make contact about the possibility of accessing the data. For example, the statement "Our access was granted through a partnership with a non-profit organization that established special agreements to facilitate this research. Researchers seeking access to the same data must engage in a similar partnership arrangement." needs to be amended to provide the details required by the data availability policy section on acceptable restrictions in relation to third-party data on this page: https://journals.plos.org/plosone/s/data-availability

Thank you for pointing at this. We have now revised the Data Availability Statement and expanded the information provided about the data. The updated version can be found below and in the cover letter. We hope this provides all the necessary information for data accessibility and replication purposes. In addition, we have included the ethical approval for data collection in the manuscript.

“This section contains information on the data collection for the study "The Effects of Remediation Programs during the COVID-19 Pandemic on the Achievements of Vulnerable Students". The study is based on data from the Netherlands Cohort Study on Education (NCO in Dutch) and information about remediation programs sourced from schools, including participation lists and content details.

Access to the NCO data is only possible via the Remote Access (RA) infrastructure of Statistics Netherlands. The data cannot be shared publicly because of privacy restrictions and cannot be exported from the secure virtual environment at Statistics Netherlands. However, all data are available at Statistics Netherlands, and researchers can request data access provided they work at an institution with which Statistics Netherlands has agreements. Requests for data access can be sent to info@nationaalcohortonderzoek.nl. A detailed description of the data and the access procedure can be found in Haelermans, C., T. Huijgen, M. Jacobs, M. Levels, R. van der Velden, L. van Vugt and S. van Wetten (2020). Using Data to Advance Educational Research, Policy and Practice: Design, Content and Research Potential of the Netherlands Cohort Study on Education. European Sociological Review, 36(4), 643-662, https://doi.org/10.1093/esr/jcaa027.

However, data on remediation programs are not openly requestable for researchers. Data obtained from funding recipients and participation lists obtained from schools are confidential and cannot be shared due to legal and ethical restrictions. Researchers seeking access for replication purposes may contact the authors and can create a project at Statistics Netherlands (provided they work at an institution with which Statistics Netherlands has agreements). The authors are allowed to provide the data on the remediation programs under strict conditions within other projects at Statistics Netherlands.”

4. Both reviewers suggest the potential the strengthen the conceptual discussion of programme characteristics and the potential benefit of using a theoretical framework to discuss their importance.

Both reviewers highlighted the need to strengthen the literature section. Reviewer 1, in remarks 4, 5, and 7, emphasised the importance of clarifying the effectiveness of interventions, specifying their target areas, and expanding the discussion on learning loss and various remediation efforts. Reviewer 2 suggested that the reported effect sizes for each intervention should be clarified and interpreted. In response to these comments, we have thoroughly reviewed and revised the entire literature section. Below, we provide a brief overview of the changes made, with more detailed responses to the reviewers' points provided separately. We give a short summary here: First, we have elaborated more on the literature related to the school closures and the learning losses, referencing several review studies that all concluded negative effects during the school closures, with worse effects for disadvantaged students. Second, for all interventions, we have elaborated on effect sizes per domain (reading and mathematics), which we also consider in this paper. These interventions are sometimes more effective in a certain domain. Third, we have incorporated potential mechanisms for why these interventions are expected to work. Lastly, we have incorporated an additional table related to all program characteristics, as well as more information about the programs in the methods section. Previously, the latter information could be found in the SI appendix; however, incorporating this into the manuscript might contribute to the discussion on effective interventions more directly.

Response to reviewer 1

Thank you for your valuable suggestions and feedback on the manuscript, and in particular your insightful comments on the literature section. In response to this comment, we have provided a more detailed description of the interventions, which we believe will improve the overall clarity of the paper. We appreciate your time and effort in reviewing our manuscript and providing constructive feedback.

• Research Question: Ensure that the research question is explicitly stated.

The research questions are now more clearly stated in the paper, specifically in the second to last paragraph of the introduction. The research questions we address in this paper are:

a) Who participated in the remediation programs? Which students were selected?

b) What is the relationship between participation in remediation programs and student achievements?

c) Which remediation programs are most effective in improving student achievement?

• Sample Size: On page 6, correct the sample size to 66,500 (not 66.500).

Thank you for pointing this out, we have now changed this.

• Literature Review Citations: In the introductory paragraph of the literature review on page 6, support claims about existing research and numerous studies with proper citations in the main text and reference section.

Thank you for this suggestion. We have now supported our claims with additional research. Several large-scale review studies, such as Donelly & Patrinos (2022), Hammerstein et al. (2021), and Betthäuser et al. (2022), have shown that students faced large losses after the COVID-19-induced school closures. To provide a comprehensive summary without having to go into detail of every individual study (which for reasons of space is not desirable, given that that is not the focus of our study), we look at review studies that examine many papers regarding initial learning losses after school closures in many countries around the world. The introductory page of the literature review now includes these and more references.

• Intervention Examples: In the second paragraph of the literature review, on page 6, list specific examples of various interventions. Clarify which interventions, such as one-on-one tutoring, are most effective.

Thank you for this suggestion. In the literature review section, we now examine various interventions, highlighting one-on-one tutoring as the most effective, supported by average effect sizes. The remainder of this section examines additional interventions, each accompanied by their respective average effect sizes. This allows for a clear comparison of the most and least effective interventions. These discussions can be found from the third paragraph onwards of the literature section.

Additionally, in relation to other points raised regarding the literature section, we have included additional studies to enhance the literature review of program characteristics. Furthermore, our answer to your remark 7 provides a detailed description of these enhancements.

• Areas of Target in Intervention: Explicitly describe the particular areas targeted by the intervention.

Thank you for this question. We would be happy to do this, however, there is not *one* intervention, but several. Hence, the short answer to this question is that it depends on the intervention. The main target area we are interested in investigating is how these interventions affect students' reading and mathematics performance. In the literature section, we have now mentioned the specific areas of language and mathematics and the expected effect sizes based on previous studies. However, some of the interventions have (additional) different target areas, that we do not focus on in this study.

Furthermore, to provide more clarity on the specific remediation programs offered, we added a table to the methodology section to show all the different elements of the programs. In the previous version of our paper, this table with descriptive statistics on the student level provides all the remediation program content could be found in the Supplementary Information (Table S3.4). This table has now been included in the main paper as Table 2. The descriptives in Table 2 show the extra efforts placed on many different remediation programs, such as on behaviour and communication or buying new methods; it also shows which goals the school set for the remediation programs, such as a language goal, mathematics goal, or study-skill goal. It also gives insight into the organisation of these programs, such as the timing, the instructors, or group sizes.

For these analyses, we have a total of N = 21,403 students included. Of these students, N = 17,408 students are within schools who have remediation programs but are not participating. These are our control group from the previous analyses. Besides, we have N = 3,959 students who participate in the remediation programs. Of these students, it differs whether they follow a program with or without a certain element. This table shows the descriptive statistics of the participating students (N = 3,959)

Table 2. Descriptive statistics about the elements of remediation programs.

% of students with element % of students without element

New methods 30.59% 69.41%

Behaviour and communication 15.91% 84.09%

Automation and guided practice 6.42% 93.58%

Remedial teaching 8.28% 91.72%

Professionalization teachers 2.73% 97.27%

Parental involvement 0.78% 99.22%

Language goal 82.29% 17.71%

Math goal 70.04% 29.96%

Study skills goal 13.08% 86.92%

Socio-emotional goal 32.81% 67.19%

Timing of remediation program

Program outside regular hours 13.08%

Program during regular hours 11.11%

Program outside & during 27.96%

Unknown 47.84%

Executor of remediation program

Internal executor(s) 42.33%

External 4.65%

Internal & external executors 50.82%

Unknown 2.20%

New or existing program

New program 36.78%

Extension of existing program 21.22%

Unknown 40.79%

Group size

Individual 2.10%

Individual and small groups 11.42%

Small groups 6.59%

Large groups 1.74%

Groups of unknown size 21.62%

Entire classes 3.71%

Unknown 52.82%

• Citations for Previous Research: The second sentence of the last paragraph on page 8 needs citations for statements related to previous research. Include corresponding references in the reference section.

Thank you. A citation has been added. Research by f.e. Lendrum & Humphrey (2012) shows that implementing interventions is important. By this, they mean that an intervention has a set of specific guidelines that must be followed to implement the intervention optimally. If this is done differently, we cannot expect the same (positive) results.

• Thematic Review of Learning Loss and Remediation: Enrich the literature review by analyzing learning loss thematically and discussing specific themes related to remediation.

Thank you for this suggestion. We have written this paper from the starting point that learning loss due to the COVID-19 pandemic is a well-established phenomenon. Several large-scale review studies have shown that students faced large losses after the COVID-19-induced school closures, for example, Donelly & Patrinos (2022), Hammerstein et al. (2021), and Betthäuser et al. (2022). To maintain focus and be mindful of the number of pages of the paper, we have briefly addressed this in the introduction rather than dedicating an extensive section of the literature review to the pandemic and the mechanisms of learning loss. Instead, we have provided a discussion on COVID-19-related learning losses in the initial two paragraphs of the literature review.

In response to your feedback, we have revised the literature review on interventions and incorporated additional literature as per remark 4. We agree with the suggestion to enhance the conceptual discussion of various interventions and programs to counter learning loss post-COVID-19 school closures. While we recognise the value of incorporating a theoretical framework, the diverse range of interventions—each with distinct pedagogical and didactical aspects—makes describing an elaborate theoretical framework for each intervention a paper in itself. Hence, we have opted not to include that as such. Instead, we have expanded the current review, offering more comprehensive information on each intervention and detailing their effectiveness in specific areas, such as reading and mathematics. These remediation programs aim to provide students requiring additional support with personalised instruction and tailored education to help them catch up. The expanded literature review explores the methods used to achieve this, including, to name a few, classroom management strategies, computer-assisted programs, and peer-tutoring.

• Method Section Variables: Enhance the method section by explicitly describing variables such as participation in the remediation program, impact of the program, and effective remediation strategies. Provide a theoretical framework for clarity.

Thank you for pointing out that this was not clear. An elaborate description of the remediation programs was part of the SI material in the previous version of the paper. The following section has been added to the manuscript: “For roughly 30,000 students, we have information about the school's program. We have the following information regarding the remediation programs: whether the remediation program purchased new methods or techniques for teaching and learning; whether the remediation program focused on behaviour and communication; whether the remediation program focused on automation and guided practice; whether the remediation program offered remedial teaching; whether the remediation program had a language goal, mathematics goal, study-skills goal, or socio-emotional goal; the timing of the program taking place outside of during normal hours; whether the program was executed by internal, external or a combination of internal and external staff; the group size of the remediation program; whether the remediation program originates from a new program or an established program.”

Additionally, we have, in relation to remark 5, provided the manuscri

---

## [Decision Letter · Decision Letter 1]

10 Dec 2024

PONE-D-24-17313R1Effective Remediation Programs for Vulnerable Students to Overcome Learning LossPLOS ONE

Dear Dr. Jacobs,

Thank you for submitting your manuscript to PLOS ONE. After careful consideration, we feel that it has merit but does not fully meet PLOS ONE’s publication criteria as it currently stands. Therefore, we invite you to submit a revised version of the manuscript that addresses the points raised during the review process.

**Thank you for submitting a revision to your article, which engaged thoroughly with the points that reviewers raised. I asked both reviewers to look at your response and revised manuscript. This does lead me to ask for some further revisions, but I would like to explain the situation as I see it further. Dealing with the two reviewers in turn:**

**Reviewer 1 has changed their assessment to Minor Revisions. Their remaining comments are fairly general and, while you are welcome to take on their suggestions, I think you have essentially satisfied their suggestions and do not propose to send the revision that I am requesting to them.**

**Reviewer 2 welcomes your strong engagement with their suggestions but, I think partly because of the improved clarity of the manuscript (which is, of course, welcome), has identified some additional concerns about aspects of the analyses for aims 2 and 3 and, so, retains a Major Revisions recommendation. I do share these concerns and, because our overriding concern should be to ensure the conclusions of the argument are well-supported by the analysis, also frame my request in this way.**

**In respect of aim 2, the reviewer flags up alternative possibilities about the potential bias of the difference in differences estimator used. My read is that such potential bias make cause incorrect conclusions to be drawn from your analysis. As such, I am keen that this is addressed if possible. Specifically, I would encourage you to explore whether the matched difference-in-differences analysis suggested is possible with the data that you have to hand (checking the plausibility of this strategy by whether you are then able to observe more encouraging parallel trends within the matched sample). However, mindful that it may not be possible with the data you have, you may alternatively submit a revision that much more clearly documents the potential biases that could arise such that the reader can very clearly judge for themselves the strength of evidence provided by this aspect of the study.**

**In respect of aim 3, it seems to me (although, as always, you have the option to rebut this in your response) that it should be possible to address and revise this aspect of the manuscript based on the further explanations and suggestions provided by reviewer 2. I would, of course, also encourage you to take on board sundry suggestions provided by the reviewer (my summary here is not exhaustive).**

**I would also like to thank you for engaging on the editorial points regarding financial disclosure and data availability. It is my assessment that these now meet the relevant policies (although the data are not publicly available you clearly document access procedures to be followed in gaining access, even if these may require moving to the Netherlands!).**

**Thank you again for engaging wholeheartedly in this process. I hope you will agree that it is helping to strengthen the robustness of the findings reported in this paper and, hence, their credibility. Although, of course, I cannot guarantee acceptance of the manuscript at this stage, I do think it will be possible with the revisions requested.**

We look forward to receiving your revised manuscript.

Kind regards,

Jake Anders

Academic Editor

PLOS ONE

Reviewers' comments:

Reviewer's Responses to Questions

**Comments to the Author**

1. If the authors have adequately addressed your comments raised in a previous round of review and you feel that this manuscript is now acceptable for publication, you may indicate that here to bypass the “Comments to the Author” section, enter your conflict of interest statement in the “Confidential to Editor” section, and submit your "Accept" recommendation.

Reviewer #1: All comments have been addressed

Reviewer #2: (No Response)

2. Is the manuscript technically sound, and do the data support the conclusions?

Reviewer #1: Yes

Reviewer #2: No

3. Has the statistical analysis been performed appropriately and rigorously? 

Reviewer #1: Yes

Reviewer #2: No

4. Have the authors made all data underlying the findings in their manuscript fully available?

Reviewer #1: Yes

Reviewer #2: No

5. Is the manuscript presented in an intelligible fashion and written in standard English?

Reviewer #1: Yes

Reviewer #2: Yes

6. Review Comments to the Author

**Reviewer #1:**  Dear Authors,

Thank you for addressing all the comments in the previous review. I have a few minor comments/suggestions this time.

You have three objectives in the introduction. I would recommend aligning these three objectives in:

(a) Literature review and theoretical framework (that needs your attention),

(b) Method (that you have done)

(c) Results (that you have done)

(d) Discussion (that needs your attention)

(e) Implication (that needs your attention).

I hope this alignment will create a clear flow of objective, process, and outcomes.

Best Regards,

Reviewer

**Reviewer #2: ** Thank you for this revised version of the manuscript. Some aspects of the manuscript have certainly improved compared to the initial version. However, in my view, some serious concerns remain concerning the second and third research aims of the paper, which I explain below.

(1) Research Aim 2

(1.1) As noted in my initial review, the divergence in learning trends between the treatment and the control group before the intervention (seen in figures S7.2-4 in the SI and discussed in the existing literature on learning losses during COVID-19 in the Netherlands), means that the current diff-in-diff research design is not suited to provide a point estimate of the true size of the effect of the remedial education programs implemented in the school year 2020-21 (Research Aim 2). But while I was initially under the impression that the DiD estimator may (only) be downwardly biased (hence my comment on the lower-bound estimate), upon reading the new version of the manuscript, it became clear that it may indeed be biased in either direction. This becomes clear when we consider that any of following could be true, given the divergent pre-trends and the unique context of the COVID-19 pandemic right before the intervention:

(1) The DID-estimator may *underestimate* the true effect of the intervention, since the treatment group (which consists of lower achieving and socio-economically disadvantaged students) may have continued to have a lower learning rate than the control group (which consists of higher achieving and socio-economically advantaged students) in the absence of the intervention, perhaps because they continued to struggle with the consequences of the COVID-19 pandemic. This is what initially led me to proposed interpreting this as a 'lower-bound' estimate may suffice. But,

(2) a second, (similarly) plausible scenario could be that the DID-estimator may *overestimate* the true effect of the intervention, since the treatment group may have recovered from the initial COVID-19 induced learning loss even in the absence of the intervention, for instance due to schools reopening in the Netherlands.

In short, due to the divergent pre-trends, the current DiD estimator could be biased in either direction, and the DiD research design does not allow for a meaningful inference on the true effect of the remedial learning interventions.

This leads me to the conclusion that the problem lies with the choice of comparison groups. The solution for approximating a more meaningful effect of the intervention may thus be to use a (propensity score) matching approach in order to compare students who are alike in terms of their previous attainment and socio-economic background, but who differ in whether they participated in the remedial education intervention. Given that the data at hand contain information on previous performance and socio-economic background this may be feasible. The question of course is how many low-performing students from low socio-economic backgrounds *did not* participate in any remedial learning intervention during the 2020-21 school year in the Netherlands. The feasibility of this alternative approach may be gauged from whether it allows for generating a control group that matches the pre-intervention learning trend during the 2019-20 school year of the treatment group. In other words, to correctly identify the effect of the remedial education intervention, one would need to work with a control group that incurred similar learning losses as the students in the treatment group, but did not receive remedial education.

I would like to emphasise that the issue of interpreting the DiD-estimator as a point estimate despite diverging pre-trends is rather serious. Many readers are likely to take the point estimate reported in the abstract (0.065 SD) at face value and conclude that the remediation programmes implement in the Netherlands after the COVID-19 pandemic were effective, even though the estimate may be strongly biased in either direction, as explained above. I hope that with the alternative approach suggested, it may be possible to address this issue. That said, if this issue cannot be addressed, it may not be possible to address the second research aim with the data at hand.

(1.2) It would be helpful to combine the first panel from figures S7.2-4 in one figure and include it in the main text (with the revised comparison groups, as proposed above). This would also help the reader gauge the substantive size of the effects. It is not clear, why Figure S7.1 is shown in the SI, given that this cohort does not form part of the analysis. For the more detailed figures in the appendix, it would be important to harmonise the scale of the y-axis across the different panels.

(2) Research Aim 3

(2.1) There seems to be a misunderstanding concerning the alternative, school-level approach that I proposed for addressing the third research aim. In their reply to the reviewer comments, the author(s) note that they "extract[ed] average student scores at the school level for the 2020/2021 academic year when the remediation programs were implemented [...] and analys[ed] the content of the remediation programs with the extracted test scores for composite, reading and mathematics" (see reply to reviewer comments p.13). But my proposal was to extract the *effect sizes* of the remedial education program for each school in the sample (rather than the average student scores) and use these as the dependent variable with program characteristics as the focal independent variables, controlling for school characteristics and characteristics of the student body participating in the remedial education program. Effect sizes could be weighed according to the precision of each estimate to account for cross-school variation herein. If certain programme characteristics would lead to remediation programmes being more effective, one should see an association between these programme characteristics and the size of the effect of the remediation programme (even if the sample size is relatively small).

Conducting the analysis both at the school and at the individual level may be a possible way to test the robustness of the findings in response to the third research aim. In this respect, it is somewhat concerning that the results from the school-level analysis suggests that programmes run by 'internal staff' are more effective (see table R1 in the response letter to the reviewer report on p.15), while the results from the individual-level analysis (shown in Figure 2b) suggests that programmes run by 'external staff' are more effective. Such contradiction may be resolved by adjusting the school-level analysis as suggested.

(2.2) As I noted in my initial report, there are two related problems concerning the individual-level analysis presented in Tables S6.1-14. The research question/aim focusses on examining "which types of remediation programs are most effective in enhancing student achievements" (p.4). In other words the aim is to compare the *relative* effectiveness of different types of remediation programmes. But the models shown in Tables S6.1-14 do not allow for inferences on the statistical significance of differences in the effectiveness between different programme types, since the reference category is students who did not participate in any remediation programme. The second problem is that the characteristics of the remedial education measures implemented in different schools are likely to co-vary and be confounded by characteristics that are not included in each of the separate models shown in Tables S6.1-13.

For their revised individual analysis, and in order to address the issue of confounding between programme characteristics, authors propose to "include these remediation program characteristics in our student-level difference-in-differences analysis". They also note that "our revised approach involves including all remediation program characteristics as control variables" (see reply to reviewer comments p.13). Since the relevant tables S6.1-14 in the SI only show 'yes' for the controls, it is not entirely clear what was done here, but it seems that only main effects were controlled for. This does not seem to solve the issue of confounding of the interaction between school-year and remediation type and also leaves the first problem noted above unsolved.

Aside from the school-level analysis proposed above, another way to address the second problem (to some extent) using an individual-level analysis might be to code the different types of remediation programmes into multiple, *mutually-exclusive* categories and run a two-way interaction between school year and this comprehensive, categorical measure of intervention type (with an additional category for no participation in any remedial program) and select one intervention type (perhaps the most common one), as the base category.

(2.3) It seems problematic, that the manuscript only selects the statistically significant results from Tables S6.1-14 for inclusion in the main text (in Figures 2a and 2b). Readers may not appreciate the possible multiple-testing bias when only seeing two out of the fourteen models that were tested in the main text. Using either of the school-level or the individual-level approach indicated above, may be one way to present the results for the different program characteristics in a simple yet encompassing way. It would be helpful to show all results from this analysis in one coefficient plot.

(2.4) It would be helpful to explain more clearly what is meant by the concepts used to describe the different program characteristics (e.g. what is meant by ‘remedial teaching’ or 'focus on language' ), as well as how they are operationalised and measured. Relatedly, it would be helpful to add more theoretical discussion of why the characteristics of different remedial education measures that are examined are thought to be relevant.

(3) Other comments:

(3.1) I agree that the current way to portray predicted probabilities in Figure 1a is similarly hard (or even harder) to interpret as the original figure using odds ratios. Using average marginal effects may allow for portraying the association btw. programme participation and different factors in a more parsimonious way while also allowing for the interpretation of the substantive size of the effect. Relatedly, it would be helpful to interpret the substantive size of the effects of different factors shown in Figure 1a on programme participation. The text currently only seems to comment on the statistical significance and the sign of coefficients, but not on their substantive size.

(3.2) Some important labels are missing in figures (e.g. y-axis labels in Figures 1a and 1b).

7. PLOS authors have the option to publish the peer review history of their article (what does this mean? ). If published, this will include your full peer review and any attached files.

**Do you want your identity to be public for this peer review?** For information about this choice, including consent withdrawal, please see our Privacy Policy .

Reviewer #1: **Yes: ** Shashidhar Belbase

Reviewer #2: No

---

## [Author Response · Author response to Decision Letter 2]

21 Feb 2025

Effective Remediation Programs for Vulnerable Students to Overcome Learning Loss

Response to reviewers

PLOSONE

February 2025

Thank you once again for your valuable feedback and suggestions. The reviewers' detailed comments and constructive criticism have significantly enhanced our paper, helping us to improve the paper’s quality and clarity. We have carefully considered each comment and incorporated the necessary revisions. In the document below, we address all the points raised by both reviewers and the editor. We truly appreciate your time and effort.

Response to the Editor

Thank you for submitting a revision to your article, which engaged thoroughly with the points that reviewers raised. I asked both reviewers to look at your response and revised manuscript. This does lead me to ask for some further revisions, but I would like to explain the situation as I see it further. Dealing with the two reviewers in turn:

Reviewer 1 has changed their assessment to Minor Revisions. Their remaining comments are fairly general and, while you are welcome to take on their suggestions, I think you have essentially satisfied their suggestions and do not propose to send the revision that I am requesting to them.

Thank you for giving us another opportunity to revise our manuscript. We have indeed taken on some of reviewer 1’s suggestions and responded to these below in our response to reviewer 1.

Reviewer 2 welcomes your strong engagement with their suggestions but, I think partly because of the improved clarity of the manuscript (which is, of course, welcome), has identified some additional concerns about aspects of the analyses for aims 2 and 3 and, so, retains a Major Revisions recommendation. I do share these concerns and, because our overriding concern should be to ensure the conclusions of the argument are well-supported by the analysis, also frame my request in this way.

Thank you. As for reviewer 2’s remaining concerns, we have included detailed explanations in our response to reviewer 2. However, we also briefly summarize these below and address some points specifically regarding reviewer 2's feedback.

In respect of aim 2, the reviewer flags up alternative possibilities about the potential bias of the difference in differences estimator used. My read is that such potential bias make cause incorrect conclusions to be drawn from your analysis. As such, I am keen that this is addressed if possible. Specifically, I would encourage you to explore whether the matched difference-in-differences analysis suggested is possible with the data that you have to hand (checking the plausibility of this strategy by whether you are then able to observe more encouraging parallel trends within the matched sample). However, mindful that it may not be possible with the data you have, you may alternatively submit a revision that much more clearly documents the potential biases that could arise such that the reader can very clearly judge for themselves the strength of evidence provided by this aspect of the study.

In response to the concern about potential biases in our results, we have explored the Difference-in-Differences analyses with propensity score matching (PSM). Furthermore, we have examined the results and pre-trends if we only keep the lower-performing half of our sample, which is the group of students on which the interventions were focusing. Based on the large similarities of these results, we have decided to include these as robustness checks in the manuscript. The results remain very robust when analyzing the DiD analyses with the PSM-matching and for the subsample of lower-performing students. Hence, we prefer the original results, as PSM-matching also imposes new assumptions and challenges, which we will elaborate on below. In the paper, we now have added an extra subsection titled "Robustness checks" in the "Results" section. We explain these two robustness checks in more detail below, in our response to reviewer 2.

In respect of aim 3, it seems to me (although, as always, you have the option to rebut this in your response) that it should be possible to address and revise this aspect of the manuscript based on the further explanations and suggestions provided by reviewer 2. I would, of course, also encourage you to take on board sundry suggestions provided by the reviewer (my summary here is not exhaustive).

We have looked into this and revised our analyses and manuscript now. We attempted to adjust our analyses based on reviewer 2's feedback to the best of our understanding in the previous round. However, based on the new feedback in the second round, we realize we apparently did not fully comprehend what reviewer 2 meant. In this round, we have once again attempted to include reviewer 2’s suggestion to perform a school-level analysis, although we must confess that we still do not fully comprehend what reviewer 2 has in mind when suggesting this type of analysis. We have tried several ways to perform a school-level analysis, but unfortunately, we keep running into problems when we try to incorporate analyses at the school level or calculate effects at the school level, regardless of our approach.

We do agree with the underlying comment that it is problematic that the characteristics of the programs are interrelated, but the suggested solution to this problem of doing a school-level analysis does not seem feasible. Therefore, we have decided to proceed with the “mutually exclusive” variable approach, as also suggested by the reviewer. We explain this in more detail below, in our response to reviewer 2.

I would also like to thank you for engaging on the editorial points regarding financial disclosure and data availability. It is my assessment that these now meet the relevant policies (although the data are not publicly available you clearly document access procedures to be followed in gaining access, even if these may require moving to the Netherlands!).

Thank you again for engaging wholeheartedly in this process. I hope you will agree that it is helping to strengthen the robustness of the findings reported in this paper and, hence, their credibility. Although, of course, I cannot guarantee acceptance of the manuscript at this stage, I do think it will be possible with the revisions requested.

Thank you! We hope our revisions are meeting your expectations!

Response to Reviewer 1

Dear Authors,

Thank you for addressing all the comments in the previous review. I have a few minor comments/suggestions this time.

You have three objectives in the introduction. I would recommend aligning these three objectives in:

(a) Literature review and theoretical framework (that needs your attention),

(b) Method (that you have done)

(c) Results (that you have done)

(d) Discussion (that needs your attention)

(e) Implication (that needs your attention).

I hope this alignment will create a clear flow of objective, process, and outcomes.

Thank you for suggesting an improved flow of the paper by better aligning the research aims of the paper in all sections of the paper. The research aims outlined in the methods section align with the three models we analyzed. Furthermore, in the results section, we have now incorporated headings to differentiate between the three aims of the paper. We have aligned the research questions in the results recap in the discussion and conclusion section. However, we have not incorporated this into the literature review and theoretical framework. This section mainly discusses literature and mechanisms related to the third research aim, discussing various remediation efforts. As we do not discuss a similar amount of literature regarding research aims 1 and 2, this section would be very much unbalanced. Hence, we decided not to use the suggested structure in this section.

Response to Reviewer 2

We especially want to thank reviewer 2 for taking the time to give such detailed and thoughtful comments. We have now incorporated these in the revised version of the paper. We hope you are satisfied with the changes that we have made.

As a short summary regarding your comments on research aim 2, based on your comments we ran additional analyses in which we conducted several PSM models, as well as reduced our sample to only the lower-performing half of our sample (which is the group of students that the interventions were focusing at) to better control for potential biases. However, we have decided to introduce these as robustness checks, as the results are very similar to our original model, as we will explain further below.

Regarding research aim 3, we have tried several ways to perform a school-level analysis, but unfortunately, we keep running into problems when we try to incorporate analyses at the school level or calculate effects at the school level, regardless of our approach. Hence, as per your suggestion, we have now created mutually exclusive variables with information about the remediation programs and used these in our analyses. A more detailed response to all your points can be found below.

Research Aim 2

(1.1) As noted in my initial review, the divergence in learning trends between the treatment and the control group before the intervention (seen in figures S7.2-4 in the SI and discussed in the existing literature on learning losses during COVID-19 in the Netherlands), means that the current diff-in-diff research design is not suited to provide a point estimate of the true size of the effect of the remedial education programs implemented in the school year 2020-21 (Research Aim 2). But while I was initially under the impression that the DiD estimator may (only) be downwardly biased (hence my comment on the lower-bound estimate), upon reading the new version of the manuscript, it became clear that it may indeed be biased in either direction. This becomes clear when we consider that any of following could be true, given the divergent pre-trends and the unique context of the COVID-19 pandemic right before the intervention:

• The DID-estimator may *underestimate* the true effect of the intervention, since the treatment group (which consists of lower achieving and socio-economically disadvantaged students) may have continued to have a lower learning rate than the control group (which consists of higher achieving and socio-economically advantaged students) in the absence of the intervention, perhaps because they continued to struggle with the consequences of the COVID-19 pandemic. This is what initially led me to proposed interpreting this as a 'lower-bound' estimate may suffice.

• But, (2) a second, (similarly) plausible scenario could be that the DID-estimator may *overestimate* the true effect of the intervention, since the treatment group may have recovered from the initial COVID-19 induced learning loss even in the absence of the intervention, for instance due to schools reopening in the Netherlands.

In short, due to the divergent pre-trends, the current DiD estimator could be biased in either direction, and the DiD research design does not allow for a meaningful inference on the true effect of the remedial learning interventions.

This leads me to the conclusion that the problem lies with the choice of comparison groups. The solution for approximating a more meaningful effect of the intervention may thus be to use a (propensity score) matching approach in order to compare students who are alike in terms of their previous attainment and socio-economic background, but who differ in whether they participated in the remedial education intervention. Given that the data at hand contain information on previous performance and socio-economic background this may be feasible. The question of course is how many low-performing students from low socio-economic backgrounds *did not* participate in any remedial learning intervention during the 2020-21 school year in the Netherlands. The feasibility of this alternative approach may be gauged from whether it allows for generating a control group that matches the pre-intervention learning trend during the 2019-20 school year of the treatment group. In other words, to correctly identify the effect of the remedial education intervention, one would need to work with a control group that incurred similar learning losses as the students in the treatment group, but did not receive remedial education.

Thank you for sharing your concerns regarding potential bias in the results of research aim 2. If we understand you correctly, your main concern regarding bias comes from the diverging pre-trends. Luckily, we do have sufficient variation in participation rate among the low-performing students from disadvantaged backgrounds. Hence, we have performed several additional analyses in which we create more balanced treatment and control groups, including your suggestion to use PSM models, with which we show very parallel pre-trends and confirm the robustness of our results. However, the methods of these additional analyses bring about new challenges, which is why we have opted to include these analyses as robustness checks rather than including (one of) them as our main analysis.

As for propensity score matching, this can indeed be used to create a more balanced treatment and control group. However, as PSM in itself ranks lower among the causality analysis methods one can employ, we have decided to combine the two techniques of DiD and PSM. The main weakness of PSM is that there might be a remaining part of the selection process due to unobservable characteristics. A DiD-model deals with this problem of unobservable background characteristics by using time-invariant fixed effects, which are similar before and after the policy introduction. Consequently, we have opted to match treatment and control groups using PSM and then generate a DiD estimator of the policy intervention. Although there are many ways to employ PSM, none come without challenges. Using Nearest-neighbor matching (1:1 or even 1:5) procedures severely reduces the sample size, which creates potential power issues. This might be problematic for research aim 2, especially for research aim 3, as we have to include many interaction terms and covariates in these regression models. And we strongly prefer to keep the same sample for all analyses in our paper. Therefore, we opted for Kernel matching, which allows us to include (almost) all observations within common support by applying Kernel weights in our regression. However, the downside of that is that we cannot use our original estimation method anymore in which we include school fixed effects (which are crucial as the interventions are organized at the school level), as varying weights per school are not allowed when using the fixed effect model (and in this case, we have varying weights per student).

We have also conducted an additional analysis to check for potential overestimation. Namely, we have conducted our main model with half the sample—the half of the students who scored lowest before the remediation programs. We also checked the parallel time trends for these students, which look more convincing. However, the downside of this method is that we lose many observations (including some of the participants, which is highly undesirable), which is problematic, especially for answering research aim 3, as we explained above.

Therefore, we have added the two additional analyses as robustness checks in our paper, to better control for potential issues that may lead to over- or underestimation of main effect. A detailed and extensive description of these two robustness checks can be found below, after your comment on Figures S7.2-4.

I would like to emphasise that the issue of interpreting the DiD-estimator as a point estimate despite diverging pre-trends is rather serious. Many readers are likely to take the point estimate reported in the abstract (0.065 SD) at face value and conclude that the remediation programmes implement in the Netherlands after the COVID-19 pandemic were effective, even though the estimate may be strongly biased in either direction, as explained above. I hope that with the alternative approach suggested, it may be possible to address this issue. That said, if this issue cannot be addressed, it may not be possible to address the second research aim with the

---

## [Decision Letter · Decision Letter 2]

18 Mar 2025

PONE-D-24-17313R2Effective Remediation Programs for Vulnerable Students to Overcome Learning LossPLOS ONE

Dear Dr. Jacobs,

Thank you for submitting your manuscript to PLOS ONE. After careful consideration, we feel that it has merit but does not fully meet PLOS ONE’s publication criteria as it currently stands. Therefore, we invite you to submit a revised version of the manuscript that addresses the points raised during the review process.

As discussed in the previous round, I have only sought a review from one of the previous reviewers so we are just ensuring that the important issues they raise have been dealt with as best as possible. Indeed, I would anticipate being able to accept the paper if you make the minor revisions noted without recourse to further review.

Ultimately, I think the biggest residual issue comes down to a disagreement over whether the parallel trends assumption is supported by the evidence provided in Figure 2, with the reviewer continuing to express concern in this regard. Looking closely at the final two points of Figure 2 I must agree that there is visible evidence of widening in more than one of the cohort/subject combinations, albeit not sizeable. Divergence in trends just ahead of treatment is, of course, potentially a particular concern because it may suggest something about the selection process into treatment being associated with pupils' trajectories.

I do agree, therefore, that it is important to caveat the findings with the risks of bias that may result from a lack of parallel trends. Ultimately, this is important regardless of the quality of pre-trends which are only suggestive since the (untestable) identifying assumption of difference in differences is that there would have been parallel trends between the treatment and comparison groups in the absence of treatment.

I, therefore, ask that you do note these (small) differences in trends just ahead of treatment in your discussion of Figure 2 and the risks that these pose for potential biases. In support of this, I support the reviewer's suggestion to reinstate the 2020/21 data into Figure 2 to help readers judge the change in gradients involved both pre- and post-treatment. I suggest including a vertical line between the pre-test and post-test periods to avoid any ambiguity. I would also say that while I like having this information in a single reference figure, following so many different patterns is a bit challenging — I would suggest more use of colours and just two patterns for the lines, perhaps variants of the same colour across Reading and Maths.

With these additions, readers will then be able to judge for themselves the quality of the evidence presented when you lay it out clearly for them with the strengths and risks.

The point made regarding the programme characteristics analyses being appropriately caveated is also important to make in this similar spirit.

I am keen to accept this paper and share the evidence that it provides but must ensure that PLOS ONE's publication criterion that the data presented in the manuscript support the conclusions drawn, which I interpret to include appropriate caveating with potential risks to its validity.

We look forward to receiving your revised manuscript.

Kind regards,

Jake Anders

Academic Editor

PLOS ONE

Journal Requirements:

Reviewers' comments:

Reviewer's Responses to Questions

**Comments to the Author**

1. If the authors have adequately addressed your comments raised in a previous round of review and you feel that this manuscript is now acceptable for publication, you may indicate that here to bypass the “Comments to the Author” section, enter your conflict of interest statement in the “Confidential to Editor” section, and submit your "Accept" recommendation.

Reviewer #2: (No Response)

2. Is the manuscript technically sound, and do the data support the conclusions?

Reviewer #2: Partly

3. Has the statistical analysis been performed appropriately and rigorously? 

Reviewer #2: Yes

4. Have the authors made all data underlying the findings in their manuscript fully available?

Reviewer #2: Yes

5. Is the manuscript presented in an intelligible fashion and written in standard English?

Reviewer #2: Yes

6. Review Comments to the Author

Reviewer #2: Thank you for this revised version of the manuscript. I appreciate all of the work that the authors have done to expand on the analyses of the initial manuscript. I also appreciate that authors engage so carefully with the feedback provided. Below I focus on an assessment of the key elements of the analysis in the current version of the manuscript: (1) the DID analysis of the overall treatment effect of the remedial learning program, and (2) the analysis of how program effects co-vary with the characteristics of specific remedial education measures chosen by different schools.

The DID analysis with both the matched sample and the subsample of low-performing students are certainly informative. However, it is clear from Figures 1-3 shown in the response to the reviewer comments, that the comparison groups are (a) not balanced on performance and, more importantly, (b) there continues to be a divergence in the pre trend between the mid-year test and the end-of-year test. I recognize that authors may feel like they have reached the end of what they can do with the data at hand and would not want to run further analyses to identify a comparison group that follows a similar the pre-trend, as the treatment group. If that is the case, I believe that it is important for the authors to be clearer about the limitations of their DID. Somewhat surprisingly, the current version of the manuscript still states that "the crucial assumption of the DiD analysis, the parallel time trend assumption, is checked and met" (p.18), when it is evidently not met. Similarly, Section of the SI still asserts that "the parallel time trend shows that both participants and non-participants move parallel over time in their developing test scores. This crucial identifying restriction is checked and shows that this assumption is not violated" (p.46). It would be important to correct this and to write more clearly when interpreting their results (p.18 onwards) that (a) the parallel trends assumption does *not* hold, (b) that results may be biased in either direction, and (c) that the effects shown should therefore be interpreted with utmost caution and further research is needed to properly identify the causal effect of the remedial intervention.

In my view it would be important to extend the timeline shown in Figures 1-3 in the response to the reviewer report (i.e. Figure 2 in the main text, and Figures S7.1-3 in the SI) to include the 2020/2021 mid-year test and the 2020/2021 end-of-year test. This would allow the reader to visually assess the substantive significance of both the pre-trend divergence and the treatment effect. Of course this would mean that these figures should be renamed from showing only 'pre-trends' to showing 'over-time change in the performance of the comparison groups'. Relatedly, while authors state that Figure 3 (in the response letter) shows "almost no diverging trends", this is of course a matter of interpretation, and showing how the performance of the comparison group continues to develop in the the 2020/2021 mid-year test and the 2020/2021 end-of-year test would allow the reader to gauge the substantive size of both the divergence in the pre-trend and the post-intervention trend.

The analyses of different program characteristics has improved as a result of the new approach of categorising and analysing these program characteristics. That said, here too, it would be important for the manuscript to highlight more clearly in the interpretation of these results (p.20-23) that the results shown constitute descriptive (as opposed to causal) evidence, that the shown associations may be confounded by unobserved heterogeneity, and that results should therefore should be interpreted with caution.

7. PLOS authors have the option to publish the peer review history of their article (what does this mean? ). If published, this will include your full peer review and any attached files.

**Do you want your identity to be public for this peer review?** For information about this choice, including consent withdrawal, please see our Privacy Policy .

Reviewer #2: No

---

## [Author Response · Author response to Decision Letter 3]

3 Apr 2025

Effective Remediation Programs for Vulnerable Students to Overcome Learning Loss

Response to reviewers

PLOSONE

April 2025

Thank you once again for the opportunity to revise the paper based on the editor’s and reviewer’s comments. We are grateful to both the editor and the reviewer for their valuable feedback, which we have incorporated into this revised version to the best of our ability. We hope it meets the expectations. In this document, we provide a detailed explanation of the changes and additions made in response to the feedback provided.

Response to the Editor

Thank you for submitting your manuscript to PLOS ONE. After careful consideration, we feel that it has merit but does not fully meet PLOS ONE’s publication criteria as it currently stands. Therefore, we invite you to submit a revised version of the manuscript that addresses the points raised during the review process. As discussed in the previous round, I have only sought a review from one of the previous reviewers so we are just ensuring that the important issues they raise have been dealt with as best as possible. Indeed, I would anticipate being able to accept the paper if you make the minor revisions noted without recourse to further review.

Thank you for giving us another opportunity to revise our manuscript.

Ultimately, I think the biggest residual issue comes down to a disagreement over whether the parallel trends assumption is supported by the evidence provided in Figure 2, with the reviewer continuing to express concern in this regard. Looking closely at the final two points of Figure 2 I must agree that there is visible evidence of widening in more than one of the cohort/subject combinations, albeit not sizeable. Divergence in trends just ahead of treatment is, of course, potentially a particular concern because it may suggest something about the selection process into treatment being associated with pupils' trajectories.

Thank you for highlighting this primary concern. We understand the concerns regarding the parallel trends and the differences between participating and non-participating students. The issues this raises about selection into treatment due to the widening gaps are, on one hand, a concern, and on the other hand, a confirmation of the right selection for the treatment group. Clearly, as schools were only permitted to establish the remediation programs for a small portion of their students, the selection was focused on those most in need of these programs. As research aim 1 shows, schools determine which students are ‘most in need’ of additional hours and support based on the test scores immediately preceding the implementation of the remediation programs. This aligns with the findings of research aim 1 and Figure 1, as well as the parallel trends, which show similarities up to the initial school closures due to COVID-19.

I do agree, therefore, that it is important to caveat the findings with the risks of bias that may result from a lack of parallel trends. Ultimately, this is important regardless of the quality of pre-trends which are only suggestive since the (untestable) identifying assumption of difference in differences is that there would have been parallel trends between the treatment and comparison groups in the absence of treatment.

Indeed, the untestable identifying assumption regarding the parallel trends, had the remediation programs not taken place, is an interesting point that has been raised. One could argue that, prior to the COVID-19 pandemic, the parallel trends were so similar that, had the treatment not occurred, the widening gaps would have persisted, as they had not been decreasing previously. However, we do not know this for sure and the parallel time trend figures are not as strong as we hoped they would be. Therefore, we have added an additional explanation to the section discussing parallel trends to mention the risk of bias due to widening gaps in the pre-trends. Below, we provide a detailed response to all additions and changes made to the figure and the manuscript.

I, therefore, ask that you do: note these (small) differences in trends just ahead of treatment in your discussion of Figure 2 and the risks that these pose for potential biases. In support of this, I support the reviewer's suggestion to reinstate the 2020/21 data into Figure 2 to help readers judge the change in gradients involved both pre- and post-treatment. I suggest including a vertical line between the pre-test and post-test periods to avoid any ambiguity. I would also say that while I like having this information in a single reference figure, following so many different patterns is a bit challenging — I would suggest more use of colours and just two patterns for the lines, perhaps variants of the same colour across Reading and Maths.

With these additions, readers will then be able to judge for themselves the quality of the evidence presented when you lay it out clearly for them with the strengths and risks.

Thank you. Let us start with the figure. We have adjusted the parallel trend figures in accordance with your suggestion, including the year 2020/2021 and the vertical line to indicate pre-trends and post-trends. Additionally, the colours and patterns have been adjusted. A solid line now always represents the participating students, and a dashed line always represents non-participating students. Cohort 1 is green, cohort 2 is blue and cohort 3 is orange. Reading always has the darker colour and mathematics has the lighter colour. Please find the updated Figure below.

Fig 2: Development of test-scores for the different cohorts in the sample.

This fig shows the parallel time trends for participants and non-participants in mathematics and reading for the three cohorts used in our analyses both before and after the remediation programs. The fig shows the pre-trends up until the end-of-year test in 2019/2020 and the post-trends in school year 2020/2021. The vertical line in the fig represents the transition from pre-trends to post-trends. The first cohort (green) started grade 1 of primary education in 2016/2017. The second cohort (blue) began in 2017/2018, and the third cohort (orange) started in 2018/2019. As reading does not have a midyear test in grade 1, the trends for reading start with the end-of-year test in grade 1.

Accompanying this figure, we discuss the pre-trends on page 18. This is a partially new paragraph, as some sentences were already present in the main text, however, we now elaborate more on the divergence in the pre-trends.

The manuscript now includes the following text: “The crucial assumption of the DiD analysis is the parallel time trend assumption (48). Fig 2 displays the time trends for the cohorts of students we include in our analyses. The results in Fig 1 have already shown that selection into the treatment is not random, rather that lower educational achievement is an important indicator for participation. In the pre-trends in Fig 2, we see divergence in progress between students who participated in the programs and non-participating students, reflecting the obvious differences in early learning losses after the first school closures, making it important to consider that the overall effects of the program might be biased in either direction (see Supporting Information section 7 for the parallel time trends).”

Additionally, the parallel trend figures for the PSM sample and half of the sample (Supporting Information Figures S7.2 and S7.3) have been updated accordingly. These are discussed in the Robustness Checks section of the manuscript, and a separate reference to these two figures is provided in the Supporting Information.

The point made regarding the programme characteristics analyses being appropriately caveated is also important to make in this similar spirit.

Thank you. We have added this in the discussion section of the manuscript. In this section, we briefly touched upon the diverging pre-trends in the manuscript, but this section has now been updated. The following paragraph can be found in the discussion now: “Given the divergence in the pre-trends regarding test score progress between the midyear test of 2019/2020 and end-of-year test in 2019/2020 among participating and non-participating students, it is important to consider that the overall effects of the program might be biased in either direction. The estimate may be biased upward if the participating students would have recovered without the remediation programs, such as due to the reopening of schools in general. Conversely, it may be biased downward if disadvantaged students, who were more likely to participate, would have continued to experience lower learning growth without the support of remediation programs. Interpreting the results in this paper as a causal effect of the remediation programs requires caution. However, this approach aims to approximate causality as good as possible, considering that these programs have a non-random treatment group. While it remains possible that our estimates are subject to both upward and downward bias, we have taken the necessary steps to strengthen their robustness. Overall, the evidence indicates that remediation programs were effective, though the precise magnitude of the effect may vary depending on the chosen specification. The same caution is needed when interpreting the estimates regarding the different characteristics of remediation programs as these estimates may be subject to bias or may be confounded by unobserved heterogeneity. Our robustness checks, including PSM models and analyses focusing on the lower-performing half of students, show somewhat more comparable pre-trends. The results of these robustness checks are largely consistent with our main outcomes.”

Journal requirements: reference list

In the current revision, no changes have been made to the manuscript's reference list. All references mentioned remain unchanged, and the reference list is complete and up to date.

Response to Reviewer 2

Reviewer #2: Thank you for this revised version of the manuscript. I appreciate all of the work that the authors have done to expand on the analyses of the initial manuscript. I also appreciate that authors engage so carefully with the feedback provided. Below I focus on an assessment of the key elements of the analysis in the current version of the manuscript: (1) the DID analysis of the overall treatment effect of the remedial learning program, and (2) the analysis of how program effects co-vary with the characteristics of specific remedial education measures chosen by different schools.

We would like to thank reviewer 2 once again for taking the time to engage with this paper and for the insightful comments provided. We hope you are satisfied with the changes we have made.

The DID analysis with both the matched sample and the subsample of low-performing students are certainly informative. However, it is clear from Figures 1-3 shown in the response to the reviewer comments, that the comparison groups are (a) not balanced on performance and, more importantly, (b) there continues to be a divergence in the pre trend between the mid-year test and the end-of-year test. I recognize that authors may feel like they have reached the end of what they can do with the data at hand and would not want to run further analyses to identify a comparison group that follows a similar the pre-trend, as the treatment group. If that is the case, I believe that it is important for the authors to be clearer about the limitations of their DID. Somewhat surprisingly, the current version of the manuscript still states that "the crucial assumption of the DiD analysis, the parallel time trend assumption, is checked and met" (p.18), when it is evidently not met. Similarly, Section of the SI still asserts that "the parallel time trend shows that both participants and non-participants move parallel over time in their developing test scores. This crucial identifying restriction is checked and shows that this assumption is not violated" (p.46). It would be important to correct this and to write more clearly when interpreting their results (p.18 onwards) that (a) the parallel trends assumption does *not* hold, (b) that results may be biased in either direction, and (c) that the effects shown should therefore be interpreted with utmost caution and further research is needed to properly identify the causal effect of the remedial intervention.

Thank you for pointing this out. Indeed, the sentence on page 18 of the manuscript and the one on page 46 in the Supporting Information should not have been phrased so strongly. This has now been revised. Additionally, the caution regarding the interpretation of the results due to the divergent trends has been addressed in the discussion of Figure 2. We have included the following text: “The crucial assumption of the DiD analysis is the parallel time trend assumption (48). Fig 2 displays the time trends for the cohorts of students we include in our analyses. The results in Fig 1 have already shown that selection into the treatment is not random, rather that lower educational achievement is an important indicator for participation. In the pre-trends in Fig 2, we see divergence in progress between students who participated in the programs and non-participating students, reflecting the obvious differences in early learning losses after the first school closures, making it important to consider that the overall effects of the program might be biased in either direction (see Supporting Information section 7 for the parallel time trends).”

Furthermore, the discussion includes a separate paragraph that specifically addresses this issue. Please find the paragraph below, following the other feedback provided.

In my view it would be important to extend the timeline shown in Figures 1-3 in the response to the reviewer report (i.e. Figure 2 in the main text, and Figures S7.1-3 in the SI) to include the 2020/2021 mid-year test and the 2020/2021 end-of-year test. This would allow the reader to visually assess the substantive significance of both the pre-trend divergence and the treatment effect. Of course this would mean that these figures should be renamed from showing only 'pre-trends' to showing 'over-time change in the performance of the comparison groups'. Relatedly, while authors state that Figure 3 (in the response letter) shows "almost no diverging trends", this is of course a matter of interpretation, and showing how the performance of the comparison group continues to develop in the the 2020/2021 mid-year test and the 2020/2021 end-of-year test would allow the reader to gauge the substantive size of both the divergence in the pre-trend and the post-intervention trend.

Thank you for your suggestions and feedback regarding Figure 2 (parallel time trends) and the recommendation to extend it. We have adjusted the parallel trend figures accordingly, including the years 2020/2021, and have added a vertical line to indicate the transition from pre-trends to post-trends. Furthermore, the colours and patterns have been modified to enhance the visibility of the different lines in the figure. In the figure, a solid line represents participating students, while a dashed line represents non-participating students. Cohort 1 is green, cohort 2 is blue, and cohort 3 is orange. Reading is depicted in the darker colour variant, while mathematics is represented in the lighter colour variant. Please find the updated figure below and in the manuscript.

The analyses of different program characteristics has improved as a result of the new approach of categorising and analysing these program characteristics. That said, here too, it would

---

## [Editor Report · Decision Letter 3]

8 Apr 2025

Effective Remediation Programs for Vulnerable Students to Overcome Learning Loss

PONE-D-24-17313R3

Dear Dr. Jacobs,

We’re pleased to inform you that your manuscript has been judged scientifically suitable for publication and will be formally accepted for publication once it meets all outstanding technical requirements.

Kind regards,

Jake Anders

Academic Editor

PLOS ONE

Additional Editor Comments (optional):

Thank you for engaging carefully with some very detailed reviewer comments. I hope you agree that the outcome has been a really valuable contribution to the literature on this topic. Well done on a great paper.
---

## [Editor Report · Acceptance letter]

PONE-D-24-17313R3

PLOS ONE

Dear Dr. Jacobs,

I'm pleased to inform you that your manuscript has been deemed suitable for publication in PLOS ONE. Congratulations! Your manuscript is now being handed over to our production team.

Kind regards,

on behalf of

Prof. Jake Anders

Academic Editor

PLOS ONE